# EFFICIENT AND MODULAR IMPLICIT DIFFERENTIATION

## ABSTRACT

Automatic differentiation (autodiff) has revolutionized machine learning. It allows expressing complex computations by composing elementary ones in creative ways and removes the burden of computing their derivatives by hand. More recently, differentiation of optimization problem solutions has attracted widespread attention with applications such as optimization layers, and in bi-level problems such as hyper-parameter optimization and meta-learning. However, so far, implicit differentiation remained difficult to use for practitioners, as it often required case-by-case tedious mathematical derivations and implementations. In this paper, we propose an efficient and modular approach for implicit differentiation of optimization problems. In our approach, the user defines directly in Python a function $F$ capturing the optimality conditions of the problem to be differentiated. Once this is done, we leverage autodiff of $F$ and implicit differentiation to automatically differentiate the optimization problem. Our approach thus combines the benefits of implicit differentiation and autodiff. It is efficient as it can be added on top of any state-of-the-art solver and modular as the optimality condition specification is decoupled from the implicit differentiation mechanism. We show that seemingly simple principles allow to recover many existing implicit differentiation methods and create new ones easily. We demonstrate the ease of formulating and solving bi-level optimization problems using our framework. We also showcase an application to the sensitivity analysis of molecular dynamics.

## 1 INTRODUCTION

Automatic differentiation (autodiff) is now an inherent part of machine learning software. It allows to express complex computations by composing elementary ones in creative ways and removes the tedious burden of computing their derivatives by hand. In parallel, the differentiation of optimization problem solutions has found many applications. A classical example is bi-level optimization, which typically involves computing the derivatives of a nested optimization problem in order to solve an outer one. Examples of applications in machine learning include hyper-parameter optimization (Chapelle et al., 2002; Seeger, 2008; Pedregosa, 2016; Franceschi et al., 2017; Bertrand et al., 2020; 2021), neural networks (Lorraine et al., 2020), and meta-learning (Franceschi et al., 2018; Rajeswaran et al., 2019). Another line of active research involving differentiation of optimization problem solutions are optimization layers (Kim et al., 2017; Amos & Kolter, 2017; Niculae & Blondel, 2017; Djolonga & Krause, 2017; Gould et al., 2019), which can be used to encourage structured outputs, and implicit deep networks (Bai et al., 2019; El Ghaoui et al., 2019), which have a smaller memory footprint than backprop-trained networks.

Since optimization problem solutions typically do not enjoy an explicit formula in terms of their inputs, autodiff cannot be used directly to differentiate these functions. In recent years, two main approaches have been developed to circumvent this problem. The first one consists of unrolling the iterations of an optimization algorithm and using the final iteration as a proxy for the optimization problem solution (Wengert, 1964; Domke, 2012; Deledalle et al., 2014; Franceschi et al., 2018; Ablin et al., 2020). This allows to **explicitly** construct a computational graph relating the algorithm output to the inputs, on which autodiff can then be used transparently. However, this requires a reimplementation of the algorithm using the autodiff system, and not all algorithms are necessarily autodiff friendly. Moreover, forward-mode autodiff has time complexity that scales linearly with the number of variables and reverse-mode autodiff has memory complexity that scales linearly with the number of algorithm iterations. In contrast, a second approach consists in **implicitly** relating an optimization problem solution to its inputs using optimality conditions. In a machine learning context, such implicit differentiation has been used for stationarity conditions (Bengio, 2000;

Lorraine et al., 2020), KKT conditions (Chapelle et al., 2002; Gould et al., 2016; Amos & Kolter, 2017; Niculae et al., 2018; Niculae & Martins, 2020) and the proximal gradient fixed point (Niculae & Blondel, 2017; Bertrand et al., 2020; 2021). An advantage of implicit differentiation is that a solver reimplementation is not needed, allowing to build upon decades of state-of-the-art software. Although implicit differentiation has a long history in numerical analysis (Griewank & Walther, 2008; Bell & Burke, 2008; Krantz & Parks, 2012; Bonnans & Shapiro, 2013), so far, it remained difficult to use for practitioners, as it required a case-by-case tedious mathematical derivation and implementation. CasADi (Andersson et al., 2019) allows to differentiate various optimization and root finding problem algorithms provided by the library. However, it does not allow to easily add implicit differentiation on top of existing solvers from optimality conditions expressed by the user, as we do. A recent tutorial explains how to implement implicit differentiation in JAX (Duvenaud et al., 2020). However, the tutorial requires the user to take care of low-level technical details and does not cover a large catalog of optimality condition mappings as we do. Other work (Agrawal et al., 2019a) attempts to address this issue by adding implicit differentiation on top of cvxpy (Diamond & Boyd, 2016). This works by reducing all convex optimization problems to a conic program and using conic programming's optimality conditions to derive an implicit differentiation formula. While this approach is very generic, solving a convex optimization problem using a conic programming solver—an ADMM-based splitting conic solver (O'Donoghue et al., 2016) in the case of cvxpy—is rarely the state-of-the-art approach for each particular problem instance.

In this work, we adopt a different strategy that makes it easy to add implicit differentiation on top of any existing solver. In our approach, the user defines directly in Python a mapping function $F$ capturing the optimality conditions of the problem solved by the algorithm. Once this is done, we leverage autodiff of $F$ combined with implicit differentiation to automatically differentiate the optimization problem solution. In this way, our approach is **generic**, yet it can exploit the **efficiency** of state-of-the-art solvers. It therefore combines the benefits of implicit differentiation and autodiff. To summarize, we make the following contributions.

- We describe our framework and its JAX implementation (provided in the supplementary material). Our framework significantly **lowers the barrier** to use implicit differentiation thanks to the use of autodiff of the optimality conditions and the seamless integration in JAX. Our framework significantly extends JAX for numerical optimization, with low-level details all abstracted away.

- We instantiate our framework on a **large catalog** of optimality conditions (Table 1), recovering existing schemes and obtaining new ones, such as the mirror descent fixed point based one.

- On the theoretical side, we provide new bounds on the **Jacobian error** when the optimization problem is only solved approximately, and empirically validate them.

- We implement four **illustrative applications**, demonstrating our framework's ease of use.

Beyond our software implementation, we hope this paper provides a **self-contained blueprint** for creating an efficient and modular implementation of implicit differentiation.

**Notation.** We denote the gradient and Hessian of $f\colon \mathbb{R}^d \to \mathbb{R}$ evaluated at $x \in \mathbb{R}^d$ by $\nabla f(x) \in \mathbb{R}^d$ and $\nabla^2 f(x) \in \mathbb{R}^{d \times d}$. We denote the Jacobian of $F\colon \mathbb{R}^d \to \mathbb{R}^p$ evaluated at $x \in \mathbb{R}^d$ by $\partial F(x) \in \mathbb{R}^{p \times d}$. When $f$ or $F$ have several arguments, we denote the gradient, Hessian and Jacobian in the $i^{\text{th}}$ argument by $\nabla_i$, $\nabla_i^2$ and $\partial_i$, respectively. The standard probability simplex is denoted by $\triangle^d := \{x \in \mathbb{R}^d\colon \|x\|_1 = 1, x \geq 0\}$. For any set $\mathcal{C} \subset \mathbb{R}^d$, we denote the indicator function $I_{\mathcal{C}}\colon \mathbb{R}^d \to \mathbb{R} \cup \{+\infty\}$ where $I_{\mathcal{C}}(x) = 0$ if $x \in \mathcal{C}$, $I_{\mathcal{C}}(x) = +\infty$ otherwise. For a vector or matrix $A$, we note $\|A\|$ the Frobenius (or Euclidean) norm, and $\|A\|_{\text{op}}$ the operator norm.

## 2 COMBINING IMPLICIT DIFFERENTIATION AND AUTODIFF

### 2.1 GENERAL PRINCIPLES

**Overview.** Contrary to autodiff through unrolled algorithm iterations, implicit differentiation typically involves a manual, sometimes complicated, mathematical derivation. For instance, numerous works (Chapelle et al., 2002; Gould et al., 2016; Amos & Kolter, 2017; Niculae et al., 2018; Niculae & Martins, 2020) use Karush–Kuhn–Tucker (KKT) conditions in order to relate a constrained optimization problem's solution to its inputs, and to manually derive a formula for its derivatives. The derivation and implementation in these works are always case-by-case.

```
X_train, y_train = load_data()  # Load features and labels

def f(x, theta):  # Objective function
  residual = jnp.dot(X_train, x) - y_train
  return (jnp.sum(residual ** 2) + theta * jnp.sum(x ** 2)) / 2

# Since f is differentiable and unconstrained, the optimality
# condition F is simply the gradient of f in the 1st argument
F = jax.grad(f, argnums=0)

@custom_root(F)
def ridge_solver(init_x, theta):
  del init_x  # Initialization not used in this solver
  XX = jnp.dot(X_train.T, X_train)
  Xy = jnp.dot(X_train.T, y_train)
  I = jnp.eye(X_train.shape[1])  # Identity matrix
  # Finds the ridge reg solution by solving a linear system
  return jnp.linalg.solve(XX + theta * I, Xy)

init_x = None
print(jax.jacobian(ridge_solver, argnums=1)(init_x, 10.0))
```

Figure 1: Adding implicit differentiation on top of a ridge regression solver. The function $f(x, \theta)$ defines the objective function and the mapping $F$, here simply equation (4), captures the optimality conditions. Our decorator `@custom_root` automatically adds implicit differentiation to the solver for the user, overriding JAX's default behavior. The last line evaluates the Jacobian at $\theta = 10$.

In this work, we propose a generic way to easily add implicit differentiation on top of existing solvers. In our approach, the user defines directly in Python a mapping function $F$ capturing the optimality conditions of the problem solved by the algorithm. We provide reusable building blocks to easily express such $F$. The provided $F$ is then plugged into our Python decorator `@custom_root`, which we append on top of the solver declaration we wish to differentiate. Under the hood, we combine implicit differentiation and autodiff of $F$ to automatically differentiate the optimization problem solution. A simple illustrative example is given in Figure 1.

**Differentiating a root.**    Let $F \colon \mathbb{R}^d \times \mathbb{R}^n \to \mathbb{R}^d$ be a user-provided mapping, capturing the optimality conditions of a problem. An optimal solution, denoted $x^\star(\theta)$, should be a **root** of $F$:
$$F(x^\star(\theta), \theta) = 0 \,. \tag{1}$$
We can see $x^\star(\theta)$ as an implicitly defined function of $\theta \in \mathbb{R}^n$, i.e., $x^\star \colon \mathbb{R}^n \to \mathbb{R}^d$. More precisely, from the **implicit function theorem** (Griewank & Walther, 2008; Krantz & Parks, 2012), we know that for $(x_0, \theta_0)$ satisfying $F(x_0, \theta_0) = 0$ with a continuously differentiable $F$, if the Jacobian $\partial_1 F$ evaluated at $(x_0, \theta_0)$ is a square invertible matrix, then there exists a function $x^\star(\cdot)$ defined on a neighborhood of $\theta_0$ such that $x^\star(\theta_0) = x_0$. Furthermore, for all $\theta$ in this neighborhood, we have that $F(x^\star(\theta), \theta) = 0$ and $\partial x^\star(\theta)$ exists. Using the chain rule, the Jacobian $\partial x^\star(\theta)$ satisfies
$$\partial_1 F(x^\star(\theta), \theta) \partial x^\star(\theta) + \partial_2 F(x^\star(\theta), \theta) = 0 \,.$$
Computing $\partial x^\star(\theta)$ therefore boils down to the resolution of the linear system of equations
$$\underbrace{-\partial_1 F(x^\star(\theta), \theta)}_{A \in \mathbb{R}^{d \times d}} \underbrace{\partial x^\star(\theta)}_{J \in \mathbb{R}^{d \times n}} = \underbrace{\partial_2 F(x^\star(\theta), \theta)}_{B \in \mathbb{R}^{d \times n}} \,. \tag{2}$$
When (1) is a one-dimensional root finding problem ($d = 1$), (2) becomes particularly simple since we then have $\nabla x^\star(\theta) = B^\top / A$, where $A$ is a scalar value.

We will show that existing and new implicit differentiation methods all reduce to this simple principle. We call our approach hybrid, since it combines implicit differentiation (and as such requires solving a linear system) with autodiff of the optimality conditions $F$. Our approach is **efficient** as it can be added on top of any state-of-the-art solver and **modular** as the optimality condition specification is **decoupled** from the implicit differentiation mechanism. This contrasts with existing works, where the mathematical derivation and implementation are specific to each optimality condition.

**Differentiating a fixed point.**    We will encounter numerous applications where $x^\star(\theta)$ is implicitly defined through a **fixed point**:
$$x^\star(\theta) = T(x^\star(\theta), \theta) \,,$$

where $T\colon \mathbb{R}^d \times \mathbb{R}^n \to \mathbb{R}^d$. This can be seen as a particular case of (1) by defining the **residual**

$$F(x, \theta) = T(x, \theta) - x\,. \tag{3}$$

In this case, using the chain rule, we have

$$A = -\partial_1 F(x^\star(\theta), \theta) = I - \partial_1 T(x^\star(\theta), \theta) \quad \text{and} \quad B = \partial_2 F(x^\star(\theta), \theta) = \partial_2 T(x^\star(\theta), \theta).$$

**Computing JVPs and VJPs.** In most practical scenarios, it is not necessary to explicitly form the Jacobian matrix, and instead it is sufficient to left-multiply or right-multiply by $\partial_1 F$ and $\partial_2 F$. These are called vector-Jacobian product (VJP) and Jacobian-vector product (JVP), and are useful for integrating $x^\star(\theta)$ with reverse-mode and forward-mode autodiff, respectively. Oftentimes, $F$ will be explicitly defined. In this case, computing the VJP or JVP can be done via autodiff. In some cases, $F$ may itself be implicitly defined, for instance when $F$ involves the solution of a variational problem. In this case, computing the VJP or JVP will itself involve implicit differentiation.

The right-multiplication (JVP) between $J = \partial x^\star(\theta)$ and a vector $v$, $Jv$, can be computed efficiently by solving $A(Jv) = Bv$. The left-multiplication (VJP) of $v^\top$ with $J$, $v^\top J$, can be computed by first solving $A^\top u = v$. Then, we can obtain $v^\top J$ by $v^\top J = u^\top A J = u^\top B$. Note that when $B$ changes but $A$ and $v$ remain the same, we do not need to solve $A^\top u = v$ once again. This allows to compute the VJP w.r.t. different variables while solving only one linear system.

To solve these linear systems, we can use the conjugate gradient method (Hestenes et al., 1952) when $A$ is symmetric positive semi-definite and GMRES (Saad & Schultz, 1986) or BiCGSTAB (Vorst & van der Vorst, 1992) otherwise. These algorithms are all matrix-free: they only require matrix-vector products. Thus, all we need from $F$ is its JVPs or VJPs. An alternative to GMRES/BiCGSTAB is to solve the normal equation $AA^\top u = Av$ using conjugate gradient. We implement this using JAX's automatic transpose routine `jax.linear_transpose` (Frostig et al., 2021).

**Pre-processing and post-processing mappings.** Oftentimes, the goal is not to differentiate $\theta$ per se, but the parameters of a function producing $\theta$. One example of such pre-processing is to convert the parameters to be differentiated from one form to another canonical form, such as a quadratic program (Amos & Kolter, 2017) or a conic program (Agrawal et al., 2019a). Another example is when $x^\star(\theta)$ is used as the output of a neural network layer, in which case $\theta$ is produced by the previous layer. Likewise, $x^\star(\theta)$ will often not be the final output we want to differentiate. One example of such post-processing is when $x^\star(\theta)$ is the solution of a dual program and we apply the dual-primal mapping to recover the solution of the primal program. Another example is the application of a loss function, in order to reduce $x^\star(\theta)$ to a scalar value. We leave the differentiation of such pre/post-processing mappings to the autodiff system, allowing to compose functions in complex ways.

**Implementation details.** When a solver function is decorated with `@custom_root`, we use `jax.custom_jvp` and `jax.custom_vjp` to automatically add custom JVP and VJP rules to the function, overriding JAX's default behavior. As mentioned above, we use linear system solvers based on matrix-vector products and therefore we only need access to $F$ through the JVP or VJP with $\partial_1 F$ and $\partial_2 F$. This is done by using `jax.jvp` and `jax.vjp`, respectively. Note that, as in Figure 1, the definition of $F$ will often include a gradient mapping $\nabla_1 f(x, \theta)$. Thankfully, JAX supports second-order derivatives transparently. For convenience, our library also provides a `@custom_fixed_point` decorator, for adding implicit differentiation on top of a solver, given a fixed point iteration $T$; see code examples in Appendix A.

## 2.2 EXAMPLES

We now give various examples of mapping $F$ or fixed point iteration $T$, recovering existing implicit differentiation methods and creating new ones. Each choice of $F$ or $T$ implies different trade-offs in terms of computational **oracles**; see Table 1. Source code examples are given in Appendix A.

**Stationary point condition.** The simplest example is to differentiate through the implicit function

$$x^\star(\theta) = \operatorname*{argmin}_{x \in \mathbb{R}^d} f(x, \theta),$$

where $f\colon \mathbb{R}^d \times \mathbb{R}^n \to \mathbb{R}$ is twice differentiable. In this case, $F$ is simply the gradient mapping

$$F(x, \theta) = \nabla_1 f(x, \theta). \tag{4}$$

Table 1: Summary of optimality condition mappings. Oracles are accessed through their JVP or VJP.

| Name | Equation | Solution needed | Oracles needed |
|------|----------|-----------------|----------------|
| Stationary | (4), (5) | Primal | $\nabla_1 f$ |
| KKT | (6) | Primal *and* dual | $\nabla_1 f, H, G, \partial_1 H, \partial_1 G$ |
| Proximal gradient | (7) | Primal | $\nabla_1 f, \mathrm{prox}_{\eta g}$ |
| Projected gradient | (9) | Primal | $\nabla_1 f, \mathrm{proj}_{\mathcal{C}}$ |
| Mirror descent | (11) | Primal | $\nabla_1 f, \mathrm{proj}_{\mathcal{C}}^{\varphi}, \nabla \varphi$ |
| Newton | (14) | Primal | $[\nabla_1^2 f(x,\theta)]^{-1}, \nabla_1 f(x,\theta)$ |
| Block proximal gradient | (15) | Primal | $[\nabla_1 f]_j, [\mathrm{prox}_{\eta g}]_j$ |
| Conic programming | (18) | Residual map root | $\mathrm{proj}_{\mathbb{R}^p \times \mathcal{K}^* \times \mathbb{R}_+}$ |

We then have $\partial_1 F(x,\theta) = \nabla_1^2 f(x,\theta)$ and $\partial_2 F(x,\theta) = \partial_2 \nabla_1 f(x,\theta)$, the Hessian of $f$ in its first argument and the Jacobian in the second argument of $\nabla_1 f(x,\theta)$. In practice, we use autodiff to compute Jacobian products automatically. Equivalently, we can use the **gradient descent fixed point**

$$T(x,\theta) = x - \eta \nabla_1 f(x,\theta), \tag{5}$$

for all $\eta > 0$. Using (3), it is easy to check that we obtain the same linear system since $\eta$ cancels out.

**KKT conditions.** We now show that the KKT conditions, manually differentiated in several works (Chapelle et al., 2002; Gould et al., 2016; Amos & Kolter, 2017; Niculae et al., 2018; Niculae & Martins, 2020), fit our framework. As we will see, the key will be to group the optimal primal and dual variables as our $x^\star(\theta)$. Let us consider the general problem

$$\underset{z \in \mathbb{R}^p}{\mathrm{argmin}}\, f(z,\theta) \quad \text{subject to} \quad G(z,\theta) \le 0,\ H(z,\theta) = 0,$$

where $z \in \mathbb{R}^p$ is the primal variable, $f\colon \mathbb{R}^p \times \mathbb{R}^n \to \mathbb{R}$, $G\colon \mathbb{R}^p \times \mathbb{R}^n \to \mathbb{R}^r$ and $H\colon \mathbb{R}^p \times \mathbb{R}^n \to \mathbb{R}^q$. The stationarity, primal feasibility and complementary slackness conditions give

$$\nabla_1 f(z,\theta) + [\partial_1 G(z,\theta)]^\top \lambda + [\partial_1 H(z,\theta)]^\top \nu = 0$$
$$H(z,\theta) = 0$$
$$\lambda \circ G(z,\theta) = 0, \tag{6}$$

where $\nu \in \mathbb{R}^q$ and $\lambda \in \mathbb{R}_+^r$ are the dual variables, also known as KKT multipliers. The primal and dual feasibility conditions can be ignored almost everywhere. The system of (potentially nonlinear) equations (6) fits our framework, as we can group the primal and dual solutions as $x^\star(\theta) = (z^\star(\theta), \nu^\star(\theta), \lambda^\star(\theta))$ to form the root of a function $F(x^\star(\theta), \theta)$, where $F\colon \mathbb{R}^d \times \mathbb{R}^n \to \mathbb{R}^d$ and $d = p + q + r$. The primal and dual solutions can be obtained from a generic solver, such as an interior point method. In practice, the above mapping $F$ will be defined directly in Python (see Figure 7 in Appendix A) and $F$ will be differentiated automatically via autodiff.

**Proximal gradient fixed point.** Unfortunately, not all algorithms return both primal and dual solutions. Moreover, if the objective contains non-smooth terms, proximal gradient descent may be more efficient. We now discuss its fixed point (Niculae & Blondel, 2017; Bertrand et al., 2020; 2021). Let $x^\star(\theta)$ be implicitly defined as

$$x^\star(\theta) := \underset{x \in \mathbb{R}^d}{\mathrm{argmin}}\, f(x,\theta) + g(x,\theta),$$

where $f\colon \mathbb{R}^d \times \mathbb{R}^n \to \mathbb{R}$ is twice-differentiable convex and $g\colon \mathbb{R}^d \times \mathbb{R}^n \to \mathbb{R}$ is convex but possibly non-smooth. Let us define the proximity operator associated with $g$ by

$$\mathrm{prox}_g(y,\theta) := \underset{x \in \mathbb{R}^d}{\mathrm{argmin}}\, \frac{1}{2}\|x - y\|_2^2 + g(x,\theta).$$

To implicitly differentiate $x^\star(\theta)$, we use the fixed point mapping (Parikh & Boyd, 2014, p.150)

$$T(x,\theta) = \mathrm{prox}_{\eta g}(x - \eta \nabla_1 f(x,\theta), \theta), \tag{7}$$

for any step size $\eta > 0$. The proximity operator is 1-Lipschitz continuous (Moreau, 1965). By Rademacher's theorem, it is differentiable almost everywhere. Many proximity operators enjoy a closed form and can easily be differentiated, as discussed in Appendix B.

**Projected gradient fixed point.** As a special case, when $g(x, \theta)$ is the indicator function $I_{\mathcal{C}(\theta)}(x)$, where $\mathcal{C}(\theta)$ is a convex set depending on $\theta$, we obtain

$$x^\star(\theta) = \underset{x \in \mathcal{C}(\theta)}{\operatorname{argmin}} f(x, \theta). \tag{8}$$

The proximity operator $\operatorname{prox}_g$ becomes the Euclidean projection onto $\mathcal{C}(\theta)$

$$\operatorname{prox}_g(y, \theta) = \operatorname{proj}_{\mathcal{C}}(y, \theta) := \underset{x \in \mathcal{C}(\theta)}{\operatorname{argmin}} \|x - y\|_2^2$$

and (7) becomes the projected gradient fixed point

$$T(x, \theta) = \operatorname{proj}_{\mathcal{C}}(x - \eta \nabla_1 f(x, \theta), \theta). \tag{9}$$

Compared to the KKT conditions, this fixed point is particularly suitable when the projection enjoys a closed form. We discuss how to compute the JVP / VJP for a wealth of convex sets in Appendix B.

**Mirror descent fixed point.** We again consider the case when $x^\star(\theta)$ is implicitly defined as the solution of (8). We now generalize the projected gradient fixed point beyond Euclidean geometry. Let the Bregman divergence $D_\varphi \colon \operatorname{dom}(\varphi) \times \operatorname{relint}(\operatorname{dom}(\varphi)) \to \mathbb{R}_+$ generated by $\varphi$ be defined by

$$D_\varphi(x, y) := \varphi(x) - \varphi(y) - \langle \nabla \varphi(y), x - y \rangle.$$

We define the Bregman projection of $y$ onto $\mathcal{C}(\theta) \subseteq \operatorname{dom}(\varphi)$ by

$$\operatorname{proj}_{\mathcal{C}}^{\varphi}(y, \theta) := \underset{x \in \mathcal{C}(\theta)}{\operatorname{argmin}} D_\varphi(x, \nabla \varphi^*(y)). \tag{10}$$

Definition (10) includes the mirror map $\nabla \varphi^*(y)$ for convenience. It can be seen as a mapping from $\mathbb{R}^d$ to $\operatorname{dom}(\varphi)$, ensuring that (10) is well-defined. The mirror descent fixed point mapping is then

$$\hat{x} = \nabla \varphi(x)$$
$$y = \hat{x} - \eta \nabla_1 f(x, \theta)$$
$$T(x, \theta) = \operatorname{proj}_{\mathcal{C}}^{\varphi}(y, \theta). \tag{11}$$

Because $T$ involves the composition of several functions, manually deriving its JVP/VJP is error prone. This shows that our approach leveraging autodiff allows to handle more advanced fixed point mappings. A common example of $\varphi$ is $\varphi(x) = \langle x, \log x - \mathbf{1} \rangle$, where $\operatorname{dom}(\varphi) = \mathbb{R}_+^d$. In this case, $D_\varphi$ is the Kullback-Leibler divergence. An advantage of the Kullback-Leibler projection is that it sometimes easier to compute than the Euclidean projection, as we detail in Appendix B.

**Other fixed points.** More fixed points are described in Appendix C.

## 3 JACOBIAN PRECISION GUARANTEES

In practice, either by the limitations of finite precision arithmetic or because we perform a finite number of iterations, we rarely reach the exact solution $x^\star(\theta)$, but instead only reach an approximate solution $\hat{x}$ and apply the implicit differentiation equation (2) at this approximate solution. This motivates the need for precision guarantees of this approach. We introduce the following formalism.

**Definition 1.** *Let $F : \mathbb{R}^d \times \mathbb{R}^n \to \mathbb{R}^d$ be an optimality criterion mapping. Let $A := -\partial_1 F$ and $B := \partial_2 F$. We define the **Jacobian estimate** at $(x, \theta)$ as the solution to the linear equation $A(x, \theta) J(x, \theta) = B(x, \theta)$. It is a function $J : \mathbb{R}^d \times \mathbb{R}^n \to \mathbb{R}^{d \times n}$.*

It holds by construction that $J(x^\star(\theta), \theta) = \partial x^\star(\theta)$. Computing $J(\hat{x}, \theta)$ for an approximate solution $\hat{x}$ of $x^\star(\theta)$ therefore allows to approximate the true Jacobian $\partial x^\star(\theta)$. In practice, an algorithm used to solve (1) depends on $\theta$. Note however that, what we compute is not the Jacobian of $\hat{x}(\theta)$, unlike works differentiating through unrolled algorithm iterations, but an estimate of $\partial x^\star(\theta)$. We therefore use the notation $\hat{x}$, leaving the dependence on $\theta$ implicit.

We develop bounds of the form $\|J(\hat{x}, \theta) - \partial x^\star(\theta)\| < C \|\hat{x} - x^\star(\theta)\|$, hence showing that the error on the estimated Jacobian is at most of the same order as that of $\hat{x}$ as an approximation of $x^\star(\theta)$. These bounds are based on the following main theorem, whose proof is included in Appendix D.

**Theorem 1** (Jacobian estimate). *Let $F : \mathbb{R}^d \times \mathbb{R}^n \to \mathbb{R}^d$. Assume that there exist $\alpha, \beta, \gamma, \varepsilon, R > 0$ such that $A = -\partial_1 F$ and $B = \partial_2 F$ satisfy, for all $v \in \mathbb{R}^d$, $\theta \in \mathbb{R}^n$ and $x$ such that $\|x - x^\star(\theta)\| \leq \varepsilon$:*

*A is well-conditioned, Lipschitz: $\|A(x, \theta)v\| \geq \alpha\|v\|$ , $\|A(x, \theta) - A(x^\star(\theta), \theta)\|_{\mathrm{op}} \leq \gamma\|x - x^\star(\theta)\|$.*

*B is bounded and Lipschitz: $\|B(x^\star(\theta), \theta)\| \leq R$ , $\|B(x, \theta) - B(x^\star(\theta), \theta)\| \leq \beta\|x - x^\star(\theta)\|$.*

*Under these conditions, when $\|\hat{x} - x^\star(\theta)\| \leq \varepsilon$, we have*

$$\|J(\hat{x}, \theta) - \partial x^\star(\theta)\| \leq \left(\beta\alpha^{-1} + \gamma R\alpha^{-2}\right)\|\hat{x} - x^\star(\theta)\|.$$

This result is inspired by (Higham, 2002, Theorem 7.2), that is concerned with the stability of solutions to inverse problems. Here we consider that $A(\cdot, \theta)$ is uniformly well-conditioned, rather than only at $x^\star(\theta)$. This does not affect the first order in $\varepsilon$ of this bound, and makes it valid for all $\hat{x}$. It is also more tailored to applications to equation-specific cases.

Indeed, Theorem 1 can be applied to specific cases. In particular, for the gradient descent fixed point, where $T(x, \theta) = x - \eta\nabla_1 f(x, \theta)$ and $F(x, \theta) = T(x, \theta) - x$, this yields

$$A(x, \theta) = \eta\nabla_1^2 f(x, \theta) \text{ and } B(x, \theta) = -\eta\partial_2\nabla_1 f(x, \theta).$$

The guarantees on Jacobian precision under regularity conditions rely on $f$ directly; see Corollary 1 in Appendix D. This reveals in particular that Jacobian estimation by implicit differentiation **gains a factor of t compared to automatic differentiation**, after $t$ iterations of gradient descent in the strongly-convex setting (Ablin et al., 2020, Proposition 3.2). While our guarantees concern the Jacobian of $x^\star(\theta)$, we note that other studies (Grazzi et al., 2020; Ji et al., 2021; Bertrand et al., 2021) give guarantees on hypergradients (i.e., the gradient of an outer objective).

We illustrate these results on ridge regression, where $x^\star(\theta) = \operatorname{argmin}_x \|\Phi x - y\|_2^2 + \sum_i \theta_i x_i^2$. This problem has the merit that the solution $x^\star(\theta)$ and its Jacobian $\partial x^\star(\theta)$ are available in closed form. By running gradient descent for $t$ iterations, we obtain an estimate $\hat{x}$ of $x^\star(\theta)$ and an estimate $J(\hat{x}, \theta)$ of $\partial x^\star(\theta)$; cf. Definition 1. By doing so for different numbers of iterations $t$, we can graph the relation between the error $\|x^\star(\theta) - \hat{x}\|_2$ and the error $\|\partial x^\star(\theta) - J(\hat{x}, \theta)\|_2$, as shown in Figure 2, empirically validating Theorem 1. The results in Figure 2 were obtained using the diabetes dataset from Efron et al. (2004), with other datasets yielding a qualitatively similar behavior. We derive similar guarantees in Corollary 2 in Appendix D for proximal gradient descent.

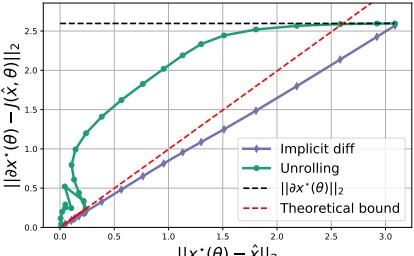

Figure 2: Jacobian estimate errors. Empirical error of implicit differentiation follows closely the theoretical upper bound. Unrolling achieves a much worse error for comparable iterate error.

## 4 EXPERIMENTS

To conclude this work, we demonstrate the ease of solving bi-level optimization problems with our framework. We also present an application to the sensitivity analysis of molecular dynamics.

### 4.1 HYPERPARAMETER OPTIMIZATION OF MULTICLASS SVMs

In this example, we consider the hyperparameter optimization of multiclass SVMs (Crammer & Singer, 2001) trained in the dual. Here, $x^\star(\theta)$ is the optimal dual solution, a matrix of shape $m \times k$, where $m$ is the number of training examples and $k$ is the number of classes, and $\theta \in \mathbb{R}_+$ is the regularization parameter. The challenge in differentiating $x^\star(\theta)$ is that each row of $x^\star(\theta)$ is constrained to belong to the probability simplex $\triangle^k$. More formally, let $X_{\mathrm{tr}} \in \mathbb{R}^{m \times p}$ be the training feature matrix and $Y_{\mathrm{tr}} \in \{0, 1\}^{m \times k}$ be the training labels (in row-wise one-hot encoding). Let $W(x, \theta) := X_{\mathrm{tr}}^\top (Y_{\mathrm{tr}} - x)/\theta \in \mathbb{R}^{p \times k}$ be the dual-primal mapping. Then, we consider the following bi-level optimization problem

$$\underbrace{\min_{\theta = \exp(\lambda)} \frac{1}{2}\|X_{\mathrm{val}}W(x^\star(\theta), \theta) - Y_{\mathrm{val}}\|_F^2}_{\text{outer problem}} \quad \text{subject to} \quad \underbrace{x^\star(\theta) = \operatorname*{argmin}_{x \in \mathcal{C}} f(x, \theta) := \frac{\theta}{2}\|W(x, \theta)\|_F^2 + \langle x, Y_{\mathrm{tr}}\rangle,}_{\text{inner problem}}$$

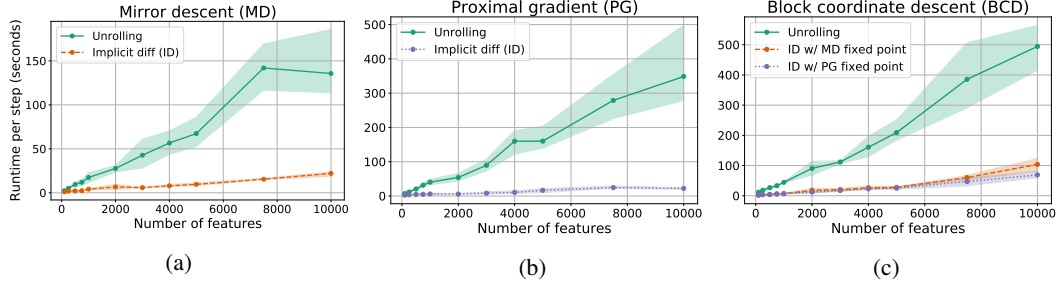

(a)     (b)     (c)

Figure 3: CPU runtime comparison of implicit differentiation and unrolling for hyperparameter optimization of multiclass SVMs for multiple problem sizes. Error bars represent 90% confidence intervals. **(a)** Mirror descent solver, with mirror descent fixed point for implicit differentiation. **(b)** Proximal gradient solver, with proximal gradient fixed point for implicit differentiation. **(c)** Block coordinate descent solver; for implicit differentiation we obtain $x^\star(\theta)$ by BCD but perform differentiation with the mirror descent and proximal gradient fixed points. This showcases that the solver and fixed point can be independently chosen.

where $\mathcal{C} = \triangle^k \times \cdots \times \triangle^k$ is the Cartesian product of $m$ probability simplices. We apply the change of variable $\theta = \exp(\lambda)$ in order to guarantee that the hyper-parameter $\theta$ is positive. The matrix $W(x^\star(\theta), \theta) \in \mathbb{R}^{p \times k}$ contains the optimal primal solution, the feature weights for each class. The outer loss is computed against validation data $X_{\text{val}}$ and $Y_{\text{val}}$.

While KKT conditions can be used to differentiate $x^\star(\theta)$, a more direct way is to use the projected gradient fixed point (9). The projection onto $\mathcal{C}$ can be easily computed by row-wise projections on the simplex. The projection's Jacobian enjoys a closed form (Appendix B). Another way to differentiate $x^\star(\theta)$ is using the mirror descent fixed point (11). Under the KL geometry, $\text{proj}_{\mathcal{C}}^\varphi(y, \theta)$ corresponds to a row-wise softmax. It is therefore easy to compute and differentiate. Figure 3 compares the runtime performance of implicit differentiation vs. unrolling for the latter two fixed points.

## 4.2 DATASET DISTILLATION

Dataset distillation (Wang et al., 2018; Lorraine et al., 2020) aims to learn a small synthetic training dataset such that a model trained on this learned data set achieves a small loss on the original training set. Formally, let $X_{\text{tr}} \in \mathbb{R}^{m \times p}$ and $y_{\text{tr}} \in [k]^m$ denote the original training set. The distilled dataset will contain one prototype example for each class and therefore $\theta \in \mathbb{R}^{k \times p}$. The dataset distillation problem can then naturally be cast as a bi-level problem, where in the inner problem we estimate a logistic regression model $x^\star(\theta) \in \mathbb{R}^{p \times k}$ trained on the distilled images $\theta \in \mathbb{R}^{k \times p}$, while in the outer problem we want to minimize the loss achieved by $x^\star(\theta)$ over the training set:

$$\underbrace{\min_{\theta \in \mathbb{R}^{k \times p}} f(x^\star(\theta), X_{\text{tr}}; y_{\text{tr}})}_{\text{outer problem}} \quad \text{subject to} \quad x^\star(\theta) \in \underbrace{\operatorname*{argmin}_{x \in \mathbb{R}^{p \times k}} f(x, \theta; [k]) + \varepsilon \|x\|^2}_{\text{inner problem}}, \quad (12)$$

where $f(W, X; y) := \ell(y, XW)$, $\ell$ denotes the multiclass logistic regression loss, and $\varepsilon = 10^{-3}$ is a regularization parameter that we found had a very positive effect on convergence.

In this problem, and unlike in the general hyperparameter optimization setup, *both* the inner and outer problems are high-dimensional, making it an ideal test-bed for gradient-based bi-level optimization methods. For this experiment, we use the MNIST dataset. The number of parameters in the inner problem is $p = 28^2 = 784$, while the number of parameters of the outer loss is $k \times p = 7840$. We solve this problem using gradient descent on both the inner and outer problem, with the gradient of the outer loss computed using implicit differentiation, as described in §2. This is fundamentally different from the approach used in the original paper, where they used differentiation of the unrolled iterates instead. For the same solver, we found that the implicit differentiation approach was 4

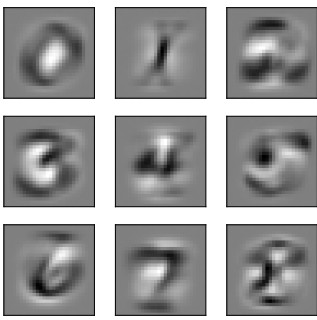

Figure 4: Distilled dataset $\theta \in \mathbb{R}^{k \times p}$ obtained by solving (12).

Table 2: Mean AUC (and 95% confidence interval) for the cancer survival prediction problem.

| Method | $L_1$ logreg | $L_2$ logreg | DictL + $L_2$ logreg | Task-driven DictL |
|---|---|---|---|---|
| AUC (%) | $71.6 \pm 2.0$ | $72.4 \pm 2.8$ | $68.3 \pm 2.3$ | $73.2 \pm 2.1$ |

times faster than the original one. The obtained distilled images $\theta$
are visualized in Figure 4.

### 4.3 Task-driven dictionary learning

Task-driven dictionary learning was proposed to learn sparse codes
for input data in such a way that the codes solve an outer learning problem (Mairal et al., 2012; Sprechmann et al., 2014; Zarka et al., 2019). Formally, given a data matrix $X_{\tt tr} \in \mathbb{R}^{m \times p}$ and a dictionary of $k$ atoms $\theta \in \mathbb{R}^{k \times p}$, a sparse code is defined as a matrix $x^\star(\theta) \in \mathbb{R}^{m \times k}$ that minimizes in $x$ a reconstruction loss $f(x, \theta) := \ell(X_{\tt tr}, x\theta)$ regularized by a sparsity-inducing penalty $g(x)$. Instead of optimizing the dictionary $\theta$ to minimize the reconstruction loss, Mairal et al. (2012) proposed to optimize an outer problem that depends on the code. Given a set of labels $Y_{\tt tr} \in \{0, 1\}^m$, we consider a logistic regression problem which results in the bilevel optimization problem:

$$\underbrace{\min_{\theta \in \mathbb{R}^{k \times p}, w \in \mathbb{R}^k, b \in \mathbb{R}} \sigma(x^\star(\theta)w + b; y_{\tt tr})}_{\text{outer problem}} \quad \text{subject to} \quad x^\star(\theta) \in \underbrace{\operatorname*{argmin}_{x \in \mathbb{R}^{m \times k}} f(x, \theta) + g(x)}_{\text{inner problem}} . \tag{13}$$

When $\ell$ is the squared Frobenius distance between matrices, and $g$ the elastic net penalty, Mairal et al. (2012, Eq. 21) derive manually, using optimality conditions (notably the support of the codes selected at the optimum), an explicit re-parameterization of $x^\star(\theta)$ as a linear system involving $\theta$. This closed-form allows for a *direct* computation of the Jacobian of $x^\star$ w.r.t. $\theta$. Similarly, (Sprechmann et al., 2014) derive first order conditions in the case where $\ell$ is a $\beta$-divergence, while (Zarka et al., 2019) propose to use unrolling of ISTA iterations. Our approach bypasses all of these manual derivations, giving the user more leisure to focus directly on modeling (loss, regularizer) aspects.

We illustrate this on breast cancer survival prediction from gene expression data. We frame it as a binary classification problem to discriminate patients who survive longer than 5 years ($m_1 = 200$) vs patients who die within 5 years of diagnosis ($m_0 = 99$), from $p = 1,000$ gene expression values. As shown in Table 2, solving (13) (Task-driven DictL) reaches a classification performance competitive with state-of-the-art $L_1$ or $L_2$ regularized logistic regression with 100 times fewer variables.

### 4.4 Sensitivity analysis of molecular dynamics

Many physical simulations require solving optimization problems, such as energy minimization in molecular (Schoenholz & Cubuk, 2020) and continuum (Beatson et al., 2020) mechanics, structural optimization (Hoyer et al., 2019) and data assimilation (Frerix et al., 2021). In this experiment, we revisit an example from JAX-MD (Schoenholz & Cubuk, 2020), the problem of finding energy minimizing configurations to a system of $k$ packed particles in a 2-dimensional box of size $\ell$

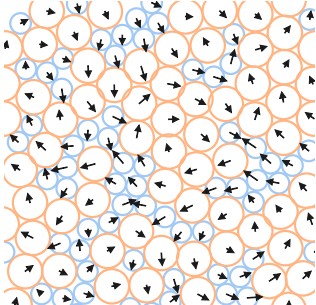

$$x^\star(\theta) = \operatorname*{argmin}_{x \in \mathbb{R}^{k \times 2}} f(x, \theta) := \sum_{i,j} U(x_{i,j}, \theta),$$

where $x^\star(\theta) \in \mathbb{R}^{k \times 2}$ are the optimal coordinates of the $k$ particles, $U(x_{i,j}, \theta)$ is the pairwise potential energy function, with half the particles at diameter 1 and half at diameter $\theta = 0.6$, which we optimize with a domain-specific optimizer (Bitzek et al., 2006). Here we consider sensitivity of particle position with respect to diameter $\partial x^\star(\theta)$, rather than sensitivity of the total energy from the original experiment. Figure 5

Figure 5: Particle positions and position sensitivity vectors, with respect to increasing the diameter of the blue particles.

shows results calculated via forward-mode implicit differentiation (JVP). Whereas differentiating the unrolled optimizer happens to work for total energy, here it typically does not even converge (see Appendix Figure 17), due the discontinuous optimization method.

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

# Appendix

## A CODE EXAMPLES

### A.1 CODE EXAMPLES FOR OPTIMALITY CONDITIONS

Our library provides several reusable optimality condition mappings $F$ or fixed points $T$. We nevertheless demonstrate the ease of writing some of them from scratch.

**Proximal gradient fixed point.** The proximal gradient fixed point (7) with step size $\eta = 1$ is $T(x, \theta) = \text{prox}_g(x - \nabla_1 f(x, \theta_f), \theta_g)$. It can be implemented as follows.

```
grad = jax.grad(f)

def T(x, theta):
  theta_f, theta_g = theta
  return prox(x - grad(x, theta_f), theta_g)
```

Figure 6: Proximal gradient fixed point $T(x, \theta)$

We recall that when the proximity operator is a projection, we recover the projected gradient fixed point as a special case. Therefore, this fixed point can also be used for constrained optimization. We provide numerous proximal and projection operators in the library.

**KKT conditions.** As a more advanced example, we now describe how to implement the KKT conditions (6). The stationarity, primal feasibility and complementary slackness conditions read

$$\nabla_1 f(z, \theta_f) + [\partial_1 G(z, \theta_G)]^\top \lambda + [\partial_1 H(z, \theta_H)]^\top \nu = 0$$
$$H(z, \theta_H) = 0$$
$$\lambda \circ G(z, \theta_G) = 0.$$

Using `jax.vjp` to compute vector-Jacobian products, this can be implemented as

```
grad = jax.grad(f)

def F(x, theta):
  z, nu, lambd = x
  theta_f, theta_H, theta_G = theta

  _, H_vjp = jax.vjp(H, z, theta_H)
  stationarity = (grad(z, theta_f) + H_vjp(nu)[0])

  primal_feasability = H(z, theta_H)

  _, G_vjp = jax.vjp(G, z, theta_G)
  stationarity += G_vjp(lambd)[0]
  comp_slackness = G(z, theta_G) * lambd

  return stationarity, primal_feasability, comp_slackness
```

Figure 7: KKT conditions $F(x, \theta)$

Similar mappings $F$ can be written if the optimization problem contains only equality constraints or only inequality constraints.

**Mirror descent fixed point.** Letting $\eta = 1$ and denoting $\theta = (\theta_f, \theta_{\text{proj}})$, the fixed point (11) is

$$\hat{x} = \nabla\varphi(x)$$
$$y = \hat{x} - \nabla_1 f(x, \theta_f)$$
$$T(x, \theta) = \text{proj}_{\mathcal{C}}^{\varphi}(y, \theta_{\text{proj}}).$$

We can then implement it as follows.

```python
grad = jax.grad(f)

def T(x, theta):
  theta_f, theta_proj = params
  x_hat = phi_mapping(x)
  y = x_hat - grad(x, theta_f)
  return bregman_projection(y, theta_proj)
```

Figure 8: Mirror descent fixed point $T(x, \theta)$

Although not considered in this example, the mapping $\nabla\varphi$ could also depend on $\theta$ if necessary.

## A.2 CODE EXAMPLES FOR EXPERIMENTS

We now sketch how to implement our experiments using our framework. In the following, `jnp` is short for `jax.numpy`. In all experiments, we only show how to compute gradients with the outer objective. We can then use these gradients with gradient-based solvers to solve the outer objective.

**Multiclass SVM experiment.**

```python
X_tr, Y_tr, X_val, Y_val = load_data()

def W(x, theta):  # dual-primal map
  return jnp.dot(X_tr.T, Y_tr - x) / theta

def f(x, theta):  # inner objective
  return (0.5 * theta * jnp.sum(W(x, theta) ** 2) +
          jnp.vdot(x, Y_tr))

grad = jax.grad(f)
proj = jax.vmap(projection_simplex)  # row-wise projections
def T(x, theta):
  return proj(x - grad(x, theta))

@custom_fixed_point(T)
def msvm_dual_solver(init_x, theta):
  # [...]
  return x_star  # solution of the dual objective

def outer_loss(lambd):
  theta = jnp.exp(lambd)
  x_star = msvm_dual_solver(init_x, theta)  # inner solution
  Y_pred = jnp.dot(W(x_star, theta), X_val)
  return 0.5 * jnp.sum((Y_pred - Y_val) ** 2)

print(jax.grad(outer_loss)(lambd))
```

Figure 9: Code example for the multiclass SVM experiment.

**Task-driven dictionary learning experiment.**

```python
X_tr, y_tr = load_data()

def f(x, theta):  # dictionary loss
  residual = X_tr - jnp.dot(x, theta)
  return huber_loss(residual)

grad = jax.grad(f)
def T(x, theta):  # proximal gradient fixed point
  return prox_lasso(x - grad(x, theta))

@custom_fixed_point(T)
def sparse_coding(init_x, theta):  # inner objective
  # [...]
  return x_star  # lasso solution

def outer_loss(theta, w):  # task-driven loss
  x_star = sparse_coding(init_x, theta)  # sparse codes
  y_pred = jnp.dot(x_star, w)
  return logloss(y_tr, y_pred)

print(jax.grad(outer_loss, argnums=(0,1)))
```

Figure 10: Code example for the task-driven dictionary learning experiment.

**Dataset distillation experiment.**

```python
X_tr, y_tr = load_data()

logloss = jax.vmap(loss.multiclass_logistic_loss)

def f(x, theta, l2reg=1e-3):  # inner objective
  scores = jnp.dot(theta, x)
  distilled_labels = jnp.arange(10)
  penalty = l2reg * jnp.sum(x * x)
  return jnp.mean(logloss(distilled_labels, scores)) + penalty

F = jax.grad(f)

@custom_root(F)
def logreg_solver(init_x, theta):
  # [...]
  return x_star

def outer_loss(theta):
  x_star = logreg_solver(init_x, theta)  # inner solution
  scores = jnp.dot(X_tr, x_star)
  return jnp.mean(logloss(y_tr, scores))

print(jax.grad(outer_loss)(theta))
```

Figure 11: Code example for the dataset distillation experiment.

**Molecular dynamics experiment.**

```
energy_fn = soft_sphere_energy_fn(diameter)
init_fn, apply_fn = jax_md.minimize.fire_descent(
    energy_fn, shift_fn)

x0 = random.uniform(key, (N, 2))
R0 = L * x0  # transform to physical coordinates
R = lax.fori_loop(
    0, num_optimization_steps,
    body_fun=lambda t, state: apply_fn(state, t=t),
    init_val=init_fn(R0)).position
x_star = R / L

def F(x, diameter):  # normalized forces
  energy_fn = soft_sphere_energy_fn(diameter)
  normalized_energy_fn = lambda x: energy_fn(L * x)
  return -jax.grad(normalized_energy_fn)(x)

dx = root_jvp(F, x_star, diameter, 1.0,
              solve=linear_solve.solve_bicgstab)

print(dx)
```

Figure 12: Code for the molecular dynamics experiment.

# B    JACOBIAN PRODUCTS

Our library provides numerous reusable building blocks. We describe in this section how to compute their Jacobian products. As a general guideline, whenever a projection enjoys a closed form, we leave the Jacobian product to the autodiff system.

## B.1    JACOBIAN PRODUCTS OF PROJECTIONS

We describe in this section how to compute the Jacobian products of the projections (in the Euclidean and KL senses) onto various convex sets. When the convex set does not depend on any variable, we simply denote it $\mathcal{C}$ instead of $\mathcal{C}(\theta)$.

**Non-negative orthant.**    When $\mathcal{C}$ is the non-negative orthant, $\mathcal{C} = \mathbb{R}_+^d$, we obtain $\text{proj}_{\mathcal{C}}(y) = \max(y, 0)$, where the maximum is evaluated element-wise. This is also known as the ReLu function. The projection in the KL sense reduces to the exponential function, $\text{proj}_{\mathcal{C}}^{\varphi}(y) = \exp(y)$.

**Box constraints.**    When $\mathcal{C}(\theta)$ is the box constraints $\mathcal{C}(\theta) = [\theta_1, \theta_2]^d$ with $\theta \in \mathbb{R}^2$, we obtain

$$\text{proj}_{\mathcal{C}}(y, \theta) = \text{clip}(y, \theta_1, \theta_2) \coloneqq \max(\min(y, \theta_2), \theta_1).$$

This is trivially extended to support different boxes for each coordinate, in which case $\theta \in \mathbb{R}^{d \times 2}$.

**Probability simplex.**    When $\mathcal{C}$ is the standard probability simplex, $\mathcal{C} = \triangle^d$, there is no analytical solution for $\text{proj}_{\mathcal{C}}(y)$. Nevertheless, the projection can be computed exactly in $O(d)$ expected time or $O(d \log d)$ worst-case time (Brucker, 1984; Michelot, 1986; Duchi et al., 2008; Condat, 2016). The Jacobian is given by $\text{diag}(s) - ss^\top / \|s\|_1$, where $s \in \{0, 1\}^d$ is a vector indicating the support of $\text{proj}_{\mathcal{C}}(y)$ (Martins & Astudillo, 2016). The projection in the KL sense, on the other hand, enjoys a closed form: it reduces to the usual softmax $\text{proj}_{\mathcal{C}}^{\varphi}(y) = \exp(y) / \sum_{j=1}^d \exp(y_j)$.

**Box sections.** Consider now the Euclidean projection $z^\star(\theta) = \text{proj}_\mathcal{C}(y, \theta)$ onto the set $\mathcal{C}(\theta) = \{z \in \mathbb{R}^d \colon \alpha_i \leq z_i \leq \beta_i, i \in [d]; w^\top z = c\}$, where $\theta = (\alpha, \beta, w, c)$. This projection is a singly-constrained bounded quadratic program. It is easy to check (see, e.g., (Niculae & Martins, 2020)) that an optimal solution satisfies for all $i \in [d]$

$$z_i^\star(\theta) = [L(x^\star(\theta), \theta)]_i := \text{clip}(w_i x^\star(\theta) + y_i, \alpha_i, \beta_i)$$

where $L \colon \mathbb{R} \times \mathbb{R}^n \to \mathbb{R}^d$ is the dual-primal mapping and $x^\star(\theta) \in \mathbb{R}$ is the optimal dual variable of the linear constraint, which should be the root of

$$F(x^\star(\theta), \theta) = L(x^\star(\theta), \theta)^\top w - c.$$

The root can be found, e.g., by bisection. The gradient $\nabla x^\star(\theta)$ is given by $\nabla x^\star(\theta) = B^\top / A$ and the Jacobian $\partial z^\star(\theta)$ is obtained by application of the chain rule on $L$.

**Norm balls.** When $\mathcal{C}(\theta) = \{x \in \mathbb{R}^d \colon \|x\| \leq \theta\}$, where $\|\cdot\|$ is a norm and $\theta \in \mathbb{R}_+$, $\text{proj}_\mathcal{C}(y, \theta)$ becomes the projection onto a norm ball. The projection onto the $\ell_1$-ball reduces to a projection onto the simplex, see, e.g., (Duchi et al., 2008). The projections onto the $\ell_2$ and $\ell_\infty$ balls enjoy a closed-form, see, e.g., (Parikh & Boyd, 2014, §6.5). Since they rely on simple composition of functions, all three projections can therefore be automatically differentiated.

**Affine sets.** When $\mathcal{C}(\theta) = \{x \in \mathbb{R}^d \colon Ax = b\}$, where $A \in \mathbb{R}^{p \times d}$, $b \in \mathbb{R}^p$ and $\theta = (A, b)$, we get

$$\text{proj}_\mathcal{C}(y, \theta) = y - A^\dagger(Ay - b) = y - A^\top(AA^\top)^{-1}(Ay - b)$$

where $A^\dagger$ is the Moore-Penrose pseudoinverse of $A$. The second equality holds if $p < d$ and $A$ is full rank. A practical implementation can pre-compute a factorization of the Gram matrix $AA^\top$. Alternatively, we can also use the KKT conditions.

**Hyperplanes and half spaces.** When $\mathcal{C}(\theta) = \{x \in \mathbb{R}^d \colon a^\top x = b\}$, where $a \in \mathbb{R}^d$ and $b \in \mathbb{R}$ and $\theta = (a, b)$, we get

$$\text{proj}_\mathcal{C}(y, \theta) = y - \frac{a^\top y - b}{\|a\|_2^2} a.$$

When $\mathcal{C}(\theta) = \{x \in \mathbb{R}^d \colon a^\top x \leq b\}$, we simply replace $a^\top y - b$ in the numerator by $\max(a^\top y - b, 0)$.

**Transportation and Birkhoff polytopes.** When $\mathcal{C}(\theta) = \{X \in \mathbb{R}^{p \times d} \colon X\mathbf{1}_d = \theta_1, X^\top \mathbf{1}_p = \theta_2, X \geq 0\}$, the so-called transportation polytope, where $\theta_1 \in \triangle^p$ and $\theta_2 \in \triangle^d$ are marginals, we can compute approximately the projections, both in the Euclidean and KL senses, by switching to the dual or semi-dual (Blondel et al., 2018). Since both are unconstrained optimization problems, we can compute their Jacobian product by implicit differentiation using the gradient descent fixed point. An advantage of the KL geometry here is that we can use Sinkhorn (Cuturi, 2013), which is a GPU-friendly algorithm. The Birkhoff polytope, the set of doubly stochastic matrices, is obtained by fixing $\theta_1 = \theta_2 = \mathbf{1}_d/d$.

**Order simplex.** When $\mathcal{C}(\theta) = \{x \in \mathbb{R}^d \colon \theta_1 \geq x_1 \geq x_2 \geq \cdots \geq x_d \geq \theta_2\}$, a so-called order simplex (Grotzinger & Witzgall, 1984; Blondel, 2019), the projection operations, both in the Euclidean and KL sense, reduce to isotonic optimization (Lim & Wright, 2016) and can be solved exactly in $O(d \log d)$ time using the Pool Adjacent Violators algorithm (Best et al., 2000). The Jacobian of the projections and efficient product with it are derived in (Djolonga & Krause, 2017; Blondel et al., 2020).

**Polyhedra.** More generally, we can consider polyhedra, i.e., sets of the form $\mathcal{C}(\theta) = \{x \in \mathbb{R}^d \colon Ax = b, Cx \leq d\}$, where $A \in \mathbb{R}^{p \times d}$, $b \in \mathbb{R}^p$, $C \in \mathbb{R}^{m \times d}$, and $d \in \mathbb{R}^m$. There are several ways to differentiate this projection. The first is to use the KKT conditions as detailed in §2.2. A second way is consider the dual of the projection instead, which is the maximization of a quadratic function subject to **non-negative constraints** (Parikh & Boyd, 2014, §6.2). That is, we can reduce the projection on a polyhedron to a problem of the form (8) with non-negative constraints, which we can in turn implicitly differentiate easily using the projected gradient fixed point, combined with the projection on the non-negative orthant. Finally, we apply the dual-primal mapping , which enjoys a closed form and is therefore amenable to autodiff, to obtain the primal projection.

### B.2 Jacobian products of proximity operators

We provide several proximity operators, including for the lasso (soft thresholding), elastic net and group lasso (block soft thresholding). All satisfy closed form expressions and can be differentiated automatically via autodiff. For more advanced proximity operators, which do not enjoy a closed form, recent works have derived their Jacobians. The Jacobians of fused lasso and OSCAR were derived in (Niculae & Blondel, 2017). For general total variation, the Jacobians were derived in (Vaiter et al., 2013; Cherkaoui et al., 2020).

## C   More examples of optimality criteria and fixed points

To demonstrate the generality of our approach, we describe in this section more optimality mapping $F$ or fixed point iteration $T$.

**Newton fixed point.**   Let $x$ be a root of $G(\cdot, \theta)$, i.e., $G(x, \theta) = 0$. The fixed point iteration of Newton's method for root-finding is

$$T(x, \theta) = x - \eta[\partial_1 G(x, \theta)]^{-1} G(x, \theta).$$

By the chain and product rules, we have

$$\partial_1 T(x, \theta) = I - \eta(...)G(x, \theta) - \eta[\partial_1 G(x, \theta)]^{-1}\partial_1 G(x, \theta) = (1 - \eta)I.$$

Using (3), we get $A = -\partial_1 F(x, \theta) = \eta I$. Similarly,

$$B = \partial_2 T(x, \theta) = \partial_2 F(x, \theta) = -\eta[\partial_1 G(x, \theta)]^{-1}\partial_2 G(x, \theta).$$

Newton's method for optimization is obtained by choosing $G(x, \theta) = \nabla_1 f(x, \theta)$, which gives

$$T(x, \theta) = x - \eta[\nabla_1^2 f(x, \theta)]^{-1}\nabla_1 f(x, \theta). \tag{14}$$

It is easy to check that we recover the same linear system as for the gradient descent fixed point (5). A practical implementation can pre-compute an LU decomposition of $\partial_1 G(x, \theta)$, or a Cholesky decomposition if $\partial_1 G(x, \theta)$ is positive semi-definite.

**Proximal block coordinate descent fixed point.**   We now consider the case when $x^\star(\theta)$ is implicitly defined as the solution

$$x^\star(\theta) \coloneqq \operatorname*{argmin}_{x \in \mathbb{R}^d} f(x, \theta) + \sum_{i=1}^m g_i(x_i, \theta),$$

where $g_1, \ldots, g_m$ are possibly non-smooth functions operating on subvectors (blocks) $x_1, \ldots, x_m$ of $x$. In this case, we can use for $i \in [m]$ the fixed point

$$x_i = [T(x, \theta)]_i = \operatorname{prox}_{\eta_i g_i}(x_i - \eta_i[\nabla_1 f(x, \theta)]_i, \theta), \tag{15}$$

where $\eta_1, \ldots, \eta_m$ are block-wise step sizes. Clearly, when the step sizes are shared, i.e., $\eta_1 = \cdots = \eta_m = \eta$, this fixed point is equivalent to the proximal gradient fixed point (7) with $g(x, \theta) = \sum_{i=1}^n g_i(x_i, \theta)$.

**Quadratic programming.**   We now show how to use the KKT conditions discussed in §2.2 to differentiate quadratic programs, recovering Optnet (Amos & Kolter, 2017) as a special case. To give some intuition, let us start with a simple equality-constrained quadratic program (QP)

$$\operatorname*{argmin}_{z \in \mathbb{R}^p} f(z, \theta) = \frac{1}{2} z^\top Q z + c^\top z \quad \text{subject to} \quad H(z, \theta) = Ez - d = 0,$$

where $Q \in \mathbb{R}^{p \times p}$, $E \in \mathbb{R}^{q \times p}$, $d \in \mathbb{R}^q$. We gather the differentiable parameters as $\theta = (Q, E, c, d)$. The stationarity and primal feasibility conditions give

$$\nabla_1 f(z, \theta) + [\partial_1 H(z, \theta)]^\top \nu = Qz + c + E^\top \nu = 0$$
$$H(z, \theta) = Ez - d = 0.$$

In matrix notation, this can be rewritten as

$$\begin{bmatrix} Q & E^\top \\ E & 0 \end{bmatrix} \begin{bmatrix} z \\ \nu \end{bmatrix} = \begin{bmatrix} -c \\ d \end{bmatrix}. \tag{16}$$

We can write the solution of the linear system (16) as a root $x = (z, \nu)$ of a function $F(x, \theta)$. More generally, the QP can also include inequality constraints

$$\underset{z \in \mathbb{R}^p}{\operatorname{argmin}} f(z, \theta) = \frac{1}{2} z^\top Q z + c^\top z \quad \text{subject to} \quad H(z, \theta) = Ez - d = 0, G(z, \theta) = Mz - h \leq 0.$$

where $M \in \mathbb{R}^{r \times p}$ and $h \in \mathbb{R}^r$. We gather the differentiable parameters as $\theta = (Q, E, M, c, d, h)$. The stationarity, primal feasibility and complementary slackness conditions give

$$\nabla_1 f(z, \theta) + [\partial_1 H(z, \theta)]^\top \nu + [\partial_1 G(z, \theta)]^\top \lambda = Qz + c + E^\top \nu + M^\top \lambda = 0$$
$$H(z, \theta) = Ez - d = 0$$
$$\lambda \circ G(z, \theta) = \operatorname{diag}(\lambda)(Mz - h) = 0$$

In matrix notation, this can be written as

$$\begin{bmatrix} Q & E^\top & M^\top \\ E & 0 & 0 \\ \operatorname{diag}(\lambda)M & 0 & 0 \end{bmatrix} \begin{bmatrix} z \\ \nu \\ \lambda \end{bmatrix} = \begin{bmatrix} -c \\ d \\ \lambda \circ h \end{bmatrix}$$

While $x = (z, \nu, \lambda)$ is no longer the solution of a linear system, it is the root of a function $F(x, \theta)$ and therefore fits our framework. With our framework, no derivation is needed. We simply define $f$, $H$ and $G$ directly in Python.

**Conic programming.** We now show that the differentiation of conic linear programs (Agrawal et al., 2019b; Amos, 2019), at the heart of differentiating through cvxpy layers (Agrawal et al., 2019a), easily fits our framework. Consider the problem

$$z^\star(\lambda), s^\star(\lambda) = \underset{z \in \mathbb{R}^p, s \in \mathbb{R}^m}{\operatorname{argmin}} c^\top z \quad \text{subject to} \quad Ez + s = d, s \in \mathcal{K}, \tag{17}$$

where $\lambda = (c, E, d)$, $E \in \mathbb{R}^{m \times p}$, $d \in \mathbb{R}^m$, $c \in \mathbb{R}^p$ and $\mathcal{K} \subseteq \mathbb{R}^m$ is a cone; $z$ and $s$ are the primal and slack variables, respectively. Every convex optimization problem can be reduced to the form (17). Let us form the skew-symmetric matrix

$$\theta(\lambda) = \begin{bmatrix} 0 & E^\top & c \\ -E & 0 & d \\ -c^\top & -d^\top & 0 \end{bmatrix} \in \mathbb{R}^{N \times N},$$

where $N = p + m + 1$. Following (Agrawal et al., 2019b;a; Amos, 2019), we can use the homogeneous self-dual embedding to reduce the process of solving (17) to finding a root of the residual map

$$F(x, \theta) = \theta \Pi x + \Pi^* x = ((\theta - I)\Pi + I)x, \tag{18}$$

where $\Pi = \operatorname{proj}_{\mathbb{R}^p \times \mathcal{K}^* \times \mathbb{R}_+}$ and $\mathcal{K}^* \subseteq \mathbb{R}^m$ is the dual cone. The splitting conic solver (O'Donoghue et al., 2016), which is based on ADMM, outputs a solution $F(x^\star(\theta), \theta) = 0$ which is decomposed as $x^\star(\theta) = (u^\star(\theta), v^\star(\theta), w^\star(\theta))$. We can then recover the optimal solution of (17) using

$$z^\star(\lambda) = u^\star(\theta(\lambda)) \quad \text{and} \quad s^\star(\lambda) = \operatorname{proj}_{\mathcal{K}^*}(v^\star(\theta(\lambda))) - v^\star(\theta(\lambda)).$$

The key oracle whose JVP/VJP we need is therefore $\Pi$, which is studied in (Ali et al., 2017). The projection onto a few cones is available in our library and can be used to express $F$.

**Frank-Wolfe.** We now consider

$$x^\star(\theta) = \underset{x \in \mathcal{C}(\theta) \subset \mathbb{R}^d}{\operatorname{argmin}} f(x, \theta), \tag{19}$$

where $\mathcal{C}(\theta)$ is a convex polytope, i.e., it is the convex hull of vertices $v_1(\theta), \ldots, v_m(\theta)$. The Frank-Wolfe algorithm requires a linear minimization oracle (LMO)

$$s \mapsto \underset{x \in \mathcal{C}(\theta)}{\operatorname{argmin}} \langle s, x \rangle$$

and is a popular algorithm when this LMO is easier to compute than the projection onto $\mathcal{C}(\theta)$. However, since this LMO is piecewise constant, its Jacobian is null almost everywhere. Inspired by SparseMAP (Niculae et al., 2018), which corresponds to the case when $f$ is a quadratic, we rewrite (19) as

$$p^\star(\theta) = \operatorname*{argmin}_{p \in \triangle^m} g(p, \theta) := f(V(\theta)p, \theta),$$

where $V(\theta)$ is a $d \times m$ matrix gathering the vertices $v_1(\theta), \ldots, v_m(\theta)$. We then have $x^\star(\theta) = V(\theta)p^\star(\theta)$. Since we have reduced (19) to minimization over the simplex, we can use the projected gradient fixed point to obtain

$$T(p^\star(\theta), \theta) = \operatorname{proj}_{\triangle^m}(p^\star(\theta) - \nabla_1 g(p^*(\theta), \theta)).$$

We can therefore compute the derivatives of $p^\star(\theta)$ by implicit differentiation and the derivatives of $x^\star(\theta)$ by product rule. Frank-Wolfe implementations typically maintain the convex weights of the vertices, which we use to get an approximation of $p^\star(\theta)$. Moreover, it is well-known that after $t$ iterations, at most $t$ vertices are visited. We can leverage this sparsity to solve a smaller linear system. Moreover, in practice, we only need to compute VJPs of $x^\star(\theta)$.

## D    JACOBIAN PRECISION PROOFS

*Proof of Theorem 1.* To simplify notations, we note $A_\star := A(x^\star, \theta)$ and $\hat{A} := A(\hat{x}, \theta)$, and similarly for $B$ and $J$. We have by definition of the Jacobian estimate function $A_\star J_\star = B_\star$ and $\hat{A}\hat{J} = \hat{B}$. Therefore we have

$$\begin{aligned}
J(\hat{x}, \theta) - \partial x^\star(\theta) &= \hat{A}^{-1}\hat{B} - A_\star^{-1}B_\star \\
&= \hat{A}^{-1}\hat{B} - \hat{A}^{-1}B_\star + \hat{A}^{-1}B_\star - A_\star^{-1}B_\star \\
&= \hat{A}^{-1}(\hat{B} - B_\star) + (\hat{A}^{-1} - A_\star^{-1})B_\star.
\end{aligned}$$

For any invertible matrices $M_1, M_2$, it holds that $M_1^{-1} - M_2^{-1} = M_1^{-1}(M_2 - M_1)M_2^{-1}$, so

$$\|M_2^{-1} - M_2^{-1}\|_{\text{op}} \le \|M_1^{-1}\|_{\text{op}}\|M_2 - M_1\|_{\text{op}}\|M_2^{-1}\|_{\text{op}}.$$

Therefore,

$$\|\hat{A}^{-1} - A_\star^{-1}\|_{\text{op}} \le \frac{1}{\alpha^2}\|\hat{A} - A_\star\|_{\text{op}} \le \frac{\gamma}{\alpha^2}\|\hat{x} - x^\star(\theta)\|.$$

As a consequence, the second term in $J(\hat{x}, \theta) - \partial x^\star(\theta)$ can be upper bounded and we obtain

$$\begin{aligned}
\|J(\hat{x}, \theta) - \partial x^\star(\theta)\| &\le \|\hat{A}^{-1}(\hat{B} - B_\star)\| + \|(\hat{A}^{-1} - A_\star^{-1})B_\star\| \\
&\le \|\hat{A}^{-1}\|_{\text{op}}\|\hat{B} - B_\star\| + \frac{\gamma}{\alpha^2}\|\hat{x} - x^\star(\theta)\|\,\|B_\star\|,
\end{aligned}$$

which yields the desired result. □

**Corollary 1** (Jacobian precision for gradient descent fixed point). *Let $f$ be such that $f(\cdot, \theta)$ is twice differentiable and $\alpha$-strongly convex and $\nabla_1^2 f(\cdot, \theta)$ is $\gamma$-Lipschitz (in the operator norm) and $\partial_2 \nabla_1 f(x, \theta)$ is $\beta$-Lipschitz and bounded in norm by $R$. The estimated Jacobian evaluated at $\hat{x}$ is then given by*

$$J(\hat{x}, \theta) = -(\nabla_1^2 f(\hat{x}, \theta))^{-1}\partial_2 \nabla_1 f(\hat{x}, \theta).$$

*For all $\theta \in \mathbb{R}^n$, and any $\hat{x}$ estimating $x^\star(\theta)$, we have the following bound for the approximation error of the estimated Jacobian*

$$\|J(\hat{x}, \theta) - \partial x^\star(\theta)\| \le \left(\frac{\beta}{\alpha} + \frac{\gamma R}{\alpha^2}\right)\|\hat{x} - x^\star(\theta)\|.$$

*Proof of Corollary 1.* This follows from Theorem 1, applied to this specific $A(x, \theta)$ and $B(x, \theta)$. □

For proximal gradient descent, where $T(x, \theta) = \operatorname{prox}_{\eta g}(x - \eta \nabla_1 f(x, \theta), \theta)$, this yields

$$\begin{aligned}
A(x, \theta) &= I - \partial_1 T(x, \theta) = I - \partial_1 \operatorname{prox}_{\eta g}(x - \eta \nabla_1 f(x, \theta), \theta)(I - \eta \nabla_1^2 f(x, \theta)) \\
B(x, \theta) &= \partial_2 \operatorname{prox}_{\eta g}(x - \eta \nabla_1 f(x, \theta), \theta) - \eta \partial_1 \operatorname{prox}_{\eta g}(x - \eta \nabla_1 f(x, \theta), \theta)\partial_2 \nabla_1 f(x, \theta).
\end{aligned}$$

We now focus in the case of proximal gradient descent on an objective $f(x, \theta) + g(x)$, where $g$ is smooth and does not depend on $\theta$. This is the case in our experiments in §4.3. Recent work also exploits local smoothness of solutions to derive similar bounds (Bertrand et al., 2021, Theorem 13)

**Corollary 2** (Jacobian precision for proximal gradient descent fixed point). *Let $f$ be such that $f(\cdot, \theta)$ is twice differentiable and $\alpha$-strongly convex and $\nabla_1^2 f(\cdot, \theta)$ is $\gamma$-Lipschitz (in the operator norm) and $\partial_2 \nabla_1 f(x, \theta)$ is $\beta$-Lipschitz and bounded in norm by $R$. Let $g : \mathbb{R}^d \to \mathbb{R}$ be a twice-differentiable $\mu$-strongly convex (with special case $\mu = 0$ being only convex), for which the function $\Gamma_\eta(x, \theta) = \nabla^2 g(prox_{\eta g}(x - \eta \nabla_1 f(x, \theta))$ is $\kappa_\eta$-Lipschitz in it first argument. The estimated Jacobian evaluated at $\hat{x}$ is then given by*

$$J(\hat{x}, \theta) = -(\nabla_1^2 f(\hat{x}, \theta) + \Gamma_\eta(\hat{x}, \theta))^{-1} \partial_2 \nabla_1 f(\hat{x}, \theta) \,.$$

*For all $\theta \in \mathbb{R}^n$, and any $\hat{x}$ estimating $x^\star(\theta)$, we have the following bound for the approximation error of the estimated Jacobian*

$$\|J(\hat{x}, \theta) - \partial x^\star(\theta)\| \leq \left( \frac{\beta + \kappa_\eta}{\alpha + \mu} + \frac{\gamma R}{(\alpha + \mu)^2} \right) \|\hat{x} - x^\star(\theta)\| \,.$$

*Proof of Corollary 2.* First, let us note that $prox_{\eta g}(y, \theta)$ does not depend on $\theta$, since $g$ itself does not depend on $\theta$, and is therefore equal to classical proximity operator of $\eta g$ which, with a slight overload of notations, we denote as $prox_{\eta g}(y)$ (with a single argument). In other words,

$$\begin{cases} prox_{\eta g}(y, \theta) & = prox_{\eta g}(y) \,, \\ \partial_1 prox_{\eta g}(y, \theta) & = \partial prox_{\eta g}(y) \,, \\ \partial_2 prox_{\eta g}(y, \theta) & = 0 \,. \end{cases}$$

Regarding the first claim (expression of the estimated Jacobian evaluated at $\hat{x}$), we first have that $prox_{\eta g}(y)$ is the solution to $(x' - y) + \eta \nabla g(x') = 0$ in $x'$ - by first-order condition for a smooth convex function. We therefore have that

$$prox_{\eta g}(y) = (I + \eta \nabla g)^{-1}(y)$$
$$\partial prox_{\eta g}(y) = (I_d + \eta \nabla^2 g(prox_{\eta g}(y)))^{-1} \,,$$

the first $I$ and inverse being functional identity and inverse, and the second $I_d$ and inverse being in the matrix sense, by inverse rule for Jacobians $\partial h(z) = [\partial h^{-1}(h(z))]^{-1}$ (applied to the prox).

As a consequence, we have, for $\Gamma_\eta(x, \theta) = \nabla^2 g(prox_{\eta g}(x - \eta \nabla_1 f(x, \theta))$ that

$$\begin{aligned} A(x, \theta) &= I_d - (I_d + \eta \Gamma_\eta(x, \theta))^{-1}(I_d - \eta \nabla_1^2 f(x, \theta)) \\ &= (I_d + \eta \Gamma_\eta(x, \theta))^{-1}[I_d + \eta \Gamma_\eta(x, \theta) - (I_d - \eta \nabla_1^2 f(x, \theta))] \\ &= \eta(I_d + \eta \Gamma_\eta(x, \theta))^{-1}(\nabla_1^2 f(x, \theta) + \Gamma_\eta(x, \theta)) \\ B(x, \theta) &= -\eta(I_d + \eta \Gamma_\eta(x, \theta))^{-1} \partial_2 \nabla_1 f(x, \theta) \,. \end{aligned}$$

As a consequence, for all $x \in \mathbb{R}^d$, we have that

$$J(x, \theta) = -(\nabla_1^2 f(x, \theta) + \Gamma_\eta(x, \theta))^{-1} \partial_2 \nabla_1 f(x, \theta) \,.$$

In the following, we modify slightly the notation of both $A$ and $B$, writing

$$\tilde{A}(x, \theta) = \nabla_1^2 f(x, \theta) + \Gamma_\eta(x, \theta)$$
$$\tilde{B}(x, \theta) = -\partial_2 \nabla_1 f(x, \theta) \,.$$

With the current hypotheses, following along the proof of Theorem 1, we have that $\tilde{A}$ is $(\alpha + \mu)$ well-conditioned, and $(\gamma + \kappa_\eta)$-Lipschitz in its first argument, and $\tilde{B}$ is $\beta$-Lipschitz in its first argument and bounded in norm by $R$. The same reasoning yields

$$\|J(\hat{x}, \theta) - \partial x^\star(\theta)\| \leq \left( \frac{\beta + \kappa_\eta}{\alpha + \mu} + \frac{\gamma R}{(\alpha + \mu)^2} \right) \|\hat{x} - x^\star(\theta)\| \,.$$

$\square$

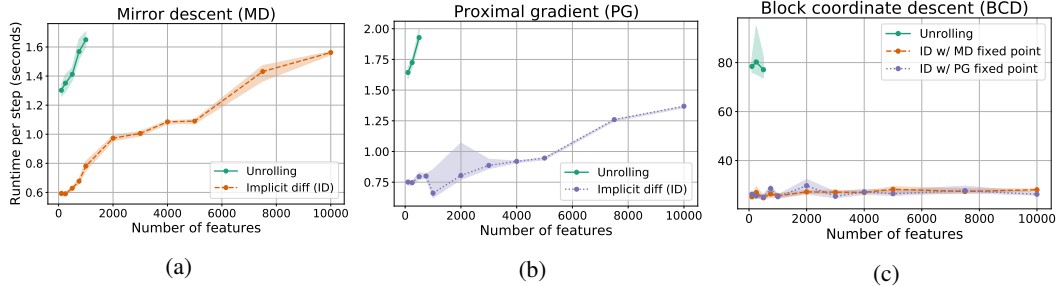

Figure 13: GPU runtime comparison of implicit differentiation and unrolling for hyperparameter optimization of multiclass SVMs for multiple problem sizes (same setting as Figure 3). Error bars represent 90% confidence intervals. Absent data points were due to out-of-memory errors (16 GB maximum).

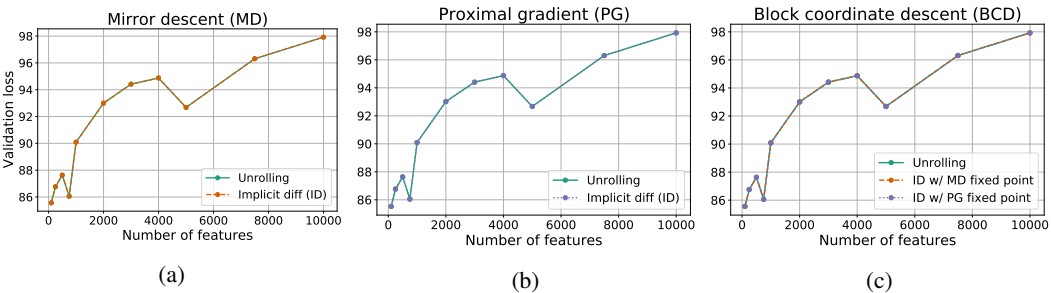

Figure 14: Value of the outer problem objective function (validation loss) for hyperparameter optimization of multiclass SVMs for multiple problem sizes (same setting as Figure 3). As can be seen, all methods performed similarly in terms of validation loss. This confirms that the faster runtimes for implicit differentiation compared to unrolling shown in Figure 3 (CPU) and Figure 13 (GPU) are not at the cost of worse validation loss.

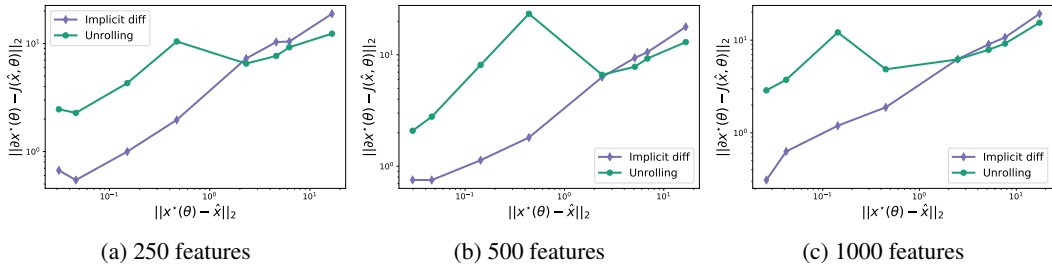

(a) 250 features  (b) 500 features  (c) 1000 features

Figure 15: Jacobian error $\|\partial x^\star(\theta) - J(\hat{x}, \theta)\|_2$ (see also Definition 1) evaluated with a regularization parameter of $\theta = 1$, as a function of solution error $\|x^\star(\theta) - \hat{x}\|_2$ when varying the number of features, on the multiclass SVM task (see Appendix E.1 for a detailed description of the experimental setup). The ground-truth solution $x^\star(\theta)$ is computed using the liblinear solver (Fan et al., 2008) available in scikit-learn (Pedregosa et al., 2011) with a very low tolerance of $10^{-9}$. Unlike in Figure 2, which was on ridge regression, the ground-truth Jacobian $\partial x^\star(\theta)$ cannot be computed in closed form, in the more difficult setting of multiclass SVMs. We therefore use a finite difference to approximately compute $\partial x^\star(\theta)$. Our results nevertheless confirm similar trends as in Figure 2.

## E EXPERIMENTAL SETUP AND ADDITIONAL RESULTS

Our experiments use JAX (Bradbury et al., 2018), which is Apache2-licensed and scikit-learn (Pedregosa et al., 2011), which is BSD-licensed.

### E.1 HYPERPARAMETER OPTIMIZATION OF MULTICLASS SVMS

**Experimental setup.** Synthetic datasets were generated using `scikit-learn`'s `sklearn.datasets.make_classification` (Pedregosa et al., 2011), following a model adapted from (Guyon, 2003). All datasets consist of $m = 700$ training samples belonging to $k = 5$ distinct classes. To simulate problems of different sizes, the number of features is varied as $p \in \{100, 250, 500, 750, 1000, 2000, 3000, 4000, 5000, 7500, 10000\}$, with 10% of features being informative and the rest random noise. In all cases, an additional $m_{\text{val}} = 200$ validation samples were generated from the same model to define the outer problem.

For the inner problem, we employed three different solvers: (i) mirror descent, (ii) (accelerated) proximal gradient descent and (iii) block coordinate descent. Hyperparameters for all solvers were individually tuned manually to ensure convergence across the range of problem sizes. For mirror descent, a stepsize of 1.0 was used for the first 100 steps, following a inverse square root decay afterwards up to a total of 2500 steps. For proximal gradient descent, a stepsize of $5 \cdot 10^{-4}$ was used for 2500 steps. The block coordinate descent solver was run for 500 iterations. All solvers used the same initialization, namely, $x_{\text{init}} = \frac{1}{k} 1_{m \times k}$, which satisfies the dual constraints.

For the outer problem, gradient descent was used with a stepsize of $5 \cdot 10^{-3}$ for the first 100 steps, following a inverse square root decay afterwards up to a total of 150 steps.

Conjugate gradient was used to solve the linear systems in implicit differentiation for at most 2500 iterations.

All results reported pertaining CPU runtimes were obtained using an internal compute cluster. GPU results were obtained using a single NVIDIA P100 GPU with 16GB of memory per dataset. For each dataset size, we report the average runtime of an individual iteration in the outer problem, alongside a 90% confidence interval estimated from the corresponding 150 runtime values.

**Additional results** Figure 13 compares the runtime of implicit differentiation and unrolling on GPU. These results highlight a fundamental limitation of the unrolling approach in memory-limited systems such as accelerators, as the inner solver suffered from out-of-memory errors for most problem sizes ($p \geq 2000$ for mirror descent, $p \geq 750$ for proximal gradient and block coordinate descent). While it might be possible to ameliorate this limitation by reducing the maximum number of iterations in the inner solver, doing so might lead to additional challenges (Wu et al., 2018) and require careful tuning.

Figure 14 depicts the validation loss (value of the outer problem objective function) at convergence. It shows that all approaches were able to solve the outer problem, with solutions produced by different approaches being qualitatively indistinguishable from each other across the range of problem sizes considered.

Figure 15 shows the Jacobian error achieved as a function of the solution error, when varying the number of features.

### E.2 TASK-DRIVEN DICTIONARY LEARNING

We downloaded from `http://acgt.cs.tau.ac.il/multi_omic_benchmark/download.html` a set of breast cancer gene expression data together with survival information generated by the TCGA Research Network (`https://www.cancer.gov/tcga`) and processed as explained by (Rappoport & Shamir, 2018). The gene expression matrix contains the expression value for p=20,531 genes in m=1,212 samples, from which we keep only the primary tumors (m=1,093). From the survival information, we select the patients who survived at least five years after diagnosis ($m_1 = 200$), and the patients who died before five years ($m_0 = 99$), resulting in a cohort of $m = 299$ patients with gene expression and binary label. Note that non-selected patients are those who are marked as alive but were not followed for 5 years.

To evaluate different binary classification methods on this cohort, we repeated 10 times a random split of the full cohort into a training (60%), validation (20%) and test (20%) sets. For each split and each method, 1) the method is trained with different parameters on the training set, 2) the parameter that maximizes the classification AUC on the validation set is selected, 3) the method is then re-trained on the union of the training and validation sets with the selected parameter, and 4) we measure the AUC of that model on the test set. We then report, for each method, the mean test AUC over the 10 repeats, together with a 95% confidence interval defined a mean $\pm 1.96 \times$ standard error of the mean.

We used Scikit Learn's implementation of logistic regression regularized by $\ell_1$ (lasso) and $\ell_2$ (ridge) penalty from `sklearn.linear_model.LogisticRegression`, and varied the `C` regularization parameter over a grid of 10 values: $\{10^{-5}, 10^{-3}, \ldots, 10^4\}$. For the unsupervised dictionary learning experiment method, we estimated a dictionary from the gene expression data in the training and validation sets, using `sklearn.decomposition.DictionaryLearning(n_components=10, alpha=2.0)`, which produces sparse codes in $k = 10$ dimensions with roughly 50% nonzero coefficients by minimizing the squared Frobenius reconstruction distance with lasso regularization on the code. We then use `sklearn.linear_model.LogisticRegression` to train a logistic regression on the codes, varying the ridge regularization parameter `C` over a grid of 10 values $\{10^{-1}, 10^0, \ldots, 10^8\}$.

Finally, we implemented the task-driven dictionary learning model (13) with our toolbox, following the pseudo-code in Figure 10. Like for the unsupervised dictionary learning experiment, we set the dimension of the codes to $k = 10$, and a fixed elastic net regularization on the inner optimization problem to ensure that the codes have roughly 50% sparsity. For the outer optimization problem, we solve an $\ell_2$ regularized ridge regression problem, varying again the ridge regularization parameter `C` over a grid of 10 values $\{10^{-1}, 10^0, \ldots, 10^8\}$. Because the outer problem is non-convex, we minimize it using the Adam optimizer (Kingma & Ba, 2014) with default parameters.

### E.3 DATASET DISTILLATION

**Experimental setup.** For the inner problem, we used gradient descent with backtracking line-search, while for the outer problem we used gradient descent with momentum and a fixed step-size. The momentum parameter was set to $0.9$ while the step-size was set to $1$.

Figure 4 was produced after 4000 iterations of the outer loop on CPU (Intel(R) Xeon(R) Platinum P-8136 CPU @ 2.00GHz), which took 1h55. Unrolled differentiation took instead 8h:05 (4 times more) to run the same number of iterations. As can be seen in Figure 16, the output is the same in both approaches.

Dataset Distillation (MNIST). Generalization Accuracy: 0.8556

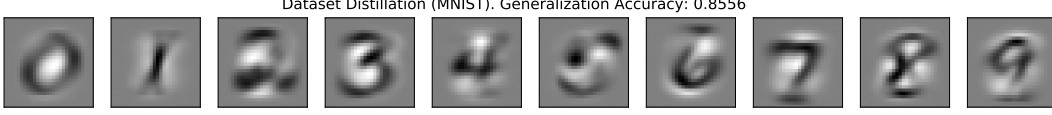

Figure 16: Distilled MNIST dataset $\theta \in \mathbb{R}^{k \times p}$ obtained by solving (12) through unrolled differentiation. Although there is no qualitative difference, the implicit differentiation approach is 4 times faster.

### E.4 MOLECULAR DYNAMICS

Our experimental setup is adapted from the JAX-MD example notebook available at https://github.com/google/jax-md/blob/master/notebooks/meta_optimization.ipynb.

We emphasize that calculating the gradient of the total energy objective, $f(x, \theta) = \sum_{ij} U(x_{i,j}, \theta)$, with respect to the diameter $\theta$ of the smaller particles, $\nabla_1 f(x, \theta)$, does not require implicit differentiation or unrolling. This is because $\nabla_1 f(x, \theta) = 0$ at $x = x^\star(\theta)$:

$$\nabla_\theta f(x^\star(\theta), \theta) = \partial x^\star(\theta)^\top \nabla_1 f(x^\star(\theta), \theta) + \nabla_2 f(x^\star(\theta), \theta) = \nabla_2 f(x^\star(\theta), \theta).$$

This is known as Danskin's theorem or envelope theorem. Thus instead, we consider sensitivities of position $\partial x^\star(\theta)$ directly, which does require implicit differentiation or unrolling.

Our results comparing implicit and unrolled differentiation for calculating the sensitivity of position are shown in Figure 17. We use BiCGSTAB (Vorst & van der Vorst, 1992) to perform the tangent linear solve. Like in the original JAX-MD experiment, we use $k = 128$ particles in $m = 2$ dimensions.

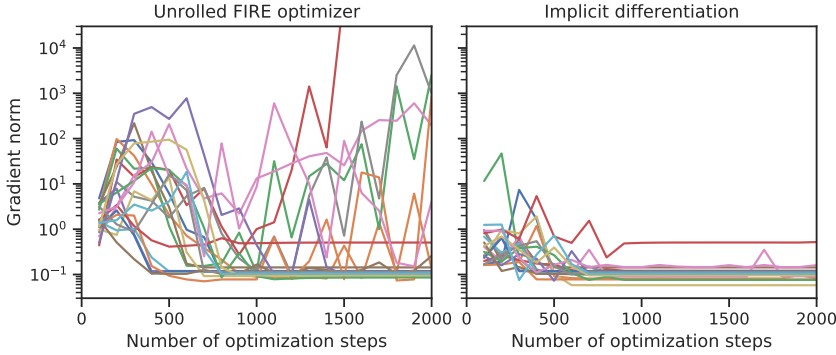

Figure 17: L1 norm of position sensitivities in the molecular dynamics simulations, for 40 different random initial conditions (different colored lines). Gradients through the unrolled FIRE optimizer (Bitzek et al., 2006) for many initial conditions do not converge, in contrast to implicit differentiation.

