# OpenReview forum: "Efficient and Modular Implicit Differentiation"
_ICLR.cc/2022/Conference — ICLR 2022 Submitted_

### Official Review · Reviewer_b1rc · 2021-10-30

**Correctness:** 3
**Technical Novelty And Significance:** 3
**Empirical Novelty And Significance:** 3
**Recommendation:** 8
**Confidence:** 3

**Main Review:**

My detailed comments are given as below.

Strength:

1. The presentation of this paper is good and most parts are easy to follow. The motivation of this paper is clear, i.e., to provide a unified and easy-to-use implementation for implicit differentiation by leveraging the tool from automatic differentiation. The tool developed here seems to be useful and cover many existing designs of interest.

2. Since implicit differentiation is useful and has critical use in practical applications now, e.g., meta-learning and automated machine learning, I feel an easy-to-use implementation can further promote the developments of more advanced algorithms. I feel this is good.

Weakness:

1. One of my concerns is the novelty of this paper. Since all things are not new, e.g., implicit differentiation, automatic differentiation, matrix-vector product computation in deep learning package, I feel this paper does not contribute a new algorithm design, but an easy-to-use package for many practical algorithms.

2. The analysis is not new as well, since there are already many works on studying the iteration complexity of the response Jacobian and the hypergradient, which I find are missing in this paper. I list some of them and highly encourage the authors to add them (and some related works therein) and provide a short discussion.

1) K. Ji, J. Yang, and Y. Liang. Bilevel optimization: Convergence analysis and enhanced design. In Proc.
International Conference on Machine Learning (ICML), 2021.

2) Grazzi, R., Franceschi, L., Pontil, M., and Salzo, S. On the iteration complexity of hypergradient computation. In Proc. International Conference on Machine Learning (ICML), 2020.

3. Experiments are conducted over only small datasets and models, e.g., synthetic data and no DNN involved. I am wondering whether this package is generalizable enough for such more practical settings. Therefore, I suggest the authors can provide some experiments on larger datasets and models.







**Summary Of The Paper:**

This paper provides a unified tool for combining the implicit differentiation technique and the  automatic differentiation method widely used in existing deep learning packages such as PyTorch and TensorFlow. The proposed implementation is easy to use for numeric optimization such as bilevel optimization, meta-learning and hyperparameter optimization because it covers many existing schemes such as fixed point, KKT point, projected method. In the experiments, the authors illustrate how their tool can be useful to simply the implementation.

**Summary Of The Review:**

Overall, I feel this work provides a useful tool for implementing the implicit differentiation in various scenarios. Although I feel the novelty of this work is not that high, I am still slightly positive about it. I am open to increase my score if the authors can address my concerns and add missing related works.

---

> ### Author Response · Authors · 2021-11-15
> **Response to reviewer b1rc**
>
> We thank the reviewer for the positive review and helpful comments.
>
> > One of my concerns is the novelty of this paper. Since all things are not new, e.g., implicit differentiation, automatic differentiation, matrix-vector product computation in deep learning package, I feel this paper does not contribute a new algorithm design, but an easy-to-use package for many practical algorithms.
>
> The implicit function theorem is of course well-known. However, we believe that our unifying perspective is new: we are not aware of any existing paper that decouples implicit differentiation from optimality conditions as we do, and that covers so many examples.
>
> > The analysis is not new as well, since there are already many works on studying the iteration complexity of the response Jacobian and the hypergradient, which I find are missing in this paper. I list some of them and highly encourage the authors to add them (and some related works therein) and provide a short discussion.
> K. Ji, J. Yang, and Y. Liang. Bilevel optimization: Convergence analysis and enhanced design. In Proc. International Conference on Machine Learning (ICML), 2021.
> Grazzi, R., Franceschi, L., Pontil, M., and Salzo, S. On the iteration complexity of hypergradient computation. In Proc. International Conference on Machine Learning (ICML), 2020.
>
> We emphasize that these two references give precision guarantees with respect to the hypergradient (i.e. the gradient of the outer objective) while we give precision guarantees with respect to the Jacobian of the inner argmin problem, in terms of the solution precision. A close inspection shows that these two references do not include similar Jacobian guarantees as an intermediate lemma either. Therefore, it is not clear how the results of these papers could easily be translated into results comparable to ours.
>
> We believe our theoretical results to be worthwhile because we can see that implicit differentiation ​​gains a factor of t compared to automatic differentiation, after t iterations of gradient descent in the strongly-convex setting (Ablin et al., 2020, Proposition 3.2). The empirical verification in Figure 2 is also insightful and novel to our knowledge.
>
> We nevertheless agree that these two references are important ones and added the following remark:
>
> While our guarantees concern the Jacobian of $x^\star(\theta)$, we note that other studies (Grazzi et al., 2020; Ji et al., 2021; Bertrand et al., 2021) give guarantees on hypergradients (i.e., the gradient of an outer objective).
>
> > Experiments are conducted over only small datasets and models, e.g., synthetic data and no DNN involved. I am wondering whether this package is generalizable enough for such more practical settings. Therefore, I suggest the authors can provide some experiments on larger datasets and models.
>
> Our software package includes an example of a deep equilibrium model (DEQ) using our framework. We show that it scales to image classification tasks such as the one in CIFAR10. This example is easily runnable and will be part of our software documentation.

---

> > ### Comment · Reviewer_b1rc · 2021-11-19
> > **Thanks for the response**
> >
> > I thank the authors for their response on the novelty and missing related works. I am satisfied with the answers and I agree that the proposed unified overview is a good point to motive such an easy-to-use package. Therefore, I increase my score to 8.
> >
> > I  have one question for my own interest. I wonder whether the developed package has implemented stochastic versions (with data sampling) of bilevel optimizers e.g.,  the implicit gradient based hypergradient estimator used in stocBiO in [1].
> >
> > [1] K. Ji, J. Yang, and Y. Liang. Bilevel optimization: Convergence analysis and enhanced design.

---

> > > ### Author Response · Authors · 2021-11-22
> > > **Response**
> > >
> > > Thanks for the constructive comments and interesting discussion. We are happy to have answered your comments positively.
> > >
> > > > I have one question for my own interest. I wonder whether the developed package has implemented stochastic versions (with data sampling) of bilevel optimizers e.g., the implicit gradient based hypergradient estimator used in stocBiO in [1].
> > >
> > > We indeed plan to add this feature in the near future.

---

### Official Review · Reviewer_h9WU · 2021-11-02

**Correctness:** 4
**Technical Novelty And Significance:** 2
**Empirical Novelty And Significance:** 2
**Recommendation:** 3
**Confidence:** 4

**Main Review:**

## Strengths

* The proposed package is a nice unification of the core implicit differentiation procedures. I believe that this construction has the potential to speed up development and application of implicit differentiation methods.
* The theorem is a good, nice-to-know theorem for the literature.
* The test tasks are nice and varied and showcase the different ways the package could be applied.

## Weaknesses

* The paper can effectively be split into the "code" portion, the "theorem" portion, and the "examples" portion. None of the portions, by themselves, meet the threshold for ICLR. In particular, the code is a package of preexisting work, and I am unaware of any similar types of packages being published to major conferences (see for example, the baseline TorchDyn library [1]). The theorem also has a straightforward, routine proof, so not much theoretical insight is gained here. Finally, the examples are smaller scale and are more toy than previous work such as [2, 3].
* The core contribution, the software, is not presented in a digestible manner. For example, the only code in the main paper is given in Figure 1, and that code is confusing. In particular, the functions $f$ and $F$ are not good names for variables, the jax.grad call operates on two arguments so should have argnums=0, and the choice to write out the functions in pure jax code clutters up the entire block. In addition, I would hope that Section 2.2 could be rewritten with more code examples. Currently, this section feels like a rehash of preexisting methods, and code would be helpful in showing how one can directly implement these in practice.

## References

[1] https://arxiv.org/abs/2009.09346

[2] https://arxiv.org/abs/1703.00443

[3] https://arxiv.org/abs/1909.01377

**Summary Of The Paper:**

This paper introduces a Jax package for implicitly differentiating various numerical solvers. Concretely, the authors develop a systemic methodology for producing gradients for a variety of optimization problems. Then, the authors prove that the Jacobian solution to the approximate numerical solution produces close enough gradients. Finally, the authors show the power of their framework on four test tasks.

**Summary Of The Review:**

I enjoyed the paper and think that the package can contribute much to popularize implicit differentiation tools. However, I feel that the paper is not sufficiently novel on any front to warrant publication. Furthermore, I would encourage the authors to present the main contribution, the package, in a more digestible manner.

---

> ### Author Response · Authors · 2021-11-15
> **Response to reviewer h9WU**
>
> > on novelty, TorchDyn, software papers
>
> We want to clarify that our paper is not just presenting a software package, but also and more importantly a unified and easy-to-use *mathematical* framework for differentiating through a variety of optimality conditions. We believe that this unifying perspective is new: we are not aware of any existing paper that decouples implicit differentiation from optimality conditions as we do, and that covers so many examples. For instance, in the Optnet paper (https://arxiv.org/abs/1703.00443), the linear system is derived manually for the KKT conditions of a quadratic program (see their equations 5 and 6). In contrast, with our approach, the KKT conditions are automatically differentiated using JAX (see our Figure 7).
>
> The TorchDyn library focuses on implicit differentiation of neural ODEs using the adjoint method, which is different from implicit differentiation of optimization problem solutions using the implicit function theorem. Both practically and conceptually, it does not serve as a baseline.
>
> An example of published paper with a combination of software and mathematical contributions is “Differentiable Convex Optimization Layers” by Agrawal et al. (NeurIPS 2019).
>
> > on straightforward proof
>
> We believe that giving guarantees on the Jacobian precision in terms of the solution precision answers a very natural question about implicit differentiation. Following the philosophy of this paper, we provided a very general result that applies to a variety of cases. For the particular case of gradient descent in the strongly-convex setting, our analysis newly shows that implicit differentiation gains a factor of $t$ after $t$ iterations, compared to autodiff (Ablin et al., 2020, Proposition 3.2). The empirical verification depicted in Figure 2 of our theoretical result is also insightful and new, to our knowledge. Simplicity of the proof and insight are in our opinion not incompatible. We feel the paper would not, in our opinion, be complete without this theoretical insight.
>
> > on "toy" examples
>
> We emphasize that the cost of implicit differentiation is mainly impacted by the dimensionality of the optimization problem solution being differentiated. With that in mind, we summarize the problem scales below.
>
> In the Optnet paper (reference [2]), the quadratic program dimensions are as follows:
> - TV denoising: 100x100 for the matrix D
> - Sudoku: 4 x 4 x 4 = 64
> - MNIST: unspecified
>
> In the DEQ paper (reference [3]), the dimensionality of the fixed point problem is that of their variable z, which is T x d, where T is the length sequence and d is the number of dimensions / number of hidden units. Since T and d are not clearly reported (except for T=400 in the copy memory task), it is difficult to judge the scale of the fixed point problem being solved and differentiated.
>
> In our experiments, the optimization problem dimensions are as follows:
> - Multiclass SVMs: n x k, where n=700 samples and k=5 classes
> - Distillation: p x k, where p=784 features and k=10 classes
> - Task-driven dictionary learning: p x k, where p = 20,000 and k=10
> - Molecular dynamics: k x 2, where k=128 particles
>
> Therefore, our largest experiment involves a dimensionality of p x k = 200,000. Moreover, we argue that our experiments are not toyish in the sense that they are varied, representative and realistic examples of implicit differentiation in a ML context.
>
> > comments on Figure 1
>
> We made the choice to call the functions f and F to match the mathematical notation in the paper. This way, the reader does not need to mentally map different notations. In actual code, we agree with you that it’s better to avoid one-letter function names.
> To answer your concern, we added more detailed comments and added `argnums=0` explicitly to jax.grad. Please notice, however, that `argnums=0` is the default behavior of jax.grad.
>
> > more code examples in Section 2.2
>
> The point of section 2.2 is to show that many different optimality conditions fall under our framework. We are not aware of an existing paper providing a unifying view of so many implicit differentiation schemes. While code examples are indeed needed, and the reviewer will find plenty of them in the appendix, they tend to take a lot of space, which is why we have prioritized other results.
>
> > I feel that the paper is not sufficiently novel on any front to warrant publication
>
> We have addressed your concerns regarding novelty and our theoretical results above, and we have pointed out an example of a NeurIPS 2019 paper, which, similarly to us, combines both software and mathematical contributions. We hope that your score can therefore reflect more  closely your enthusiasm on some aspects of the paper (“The proposed package is a nice unification of the core implicit differentiation procedures”, “The theorem is a good, nice-to-know theorem for the literature”, “The test tasks are nice and varied and showcase the different ways the package could be applied”).

---

> > ### Comment · Reviewer_h9WU · 2021-11-19
> > **Response to Response**
> >
> > Some of my comments need to be clarified.
> >
> > ## Clarification about Software Paper
> >
> > There appears to be some confusion about what I meant here. My criticism is not that the paper presents a software package, as software is an integral part of the research environment. Rather, since the main goal of this work is to streamline existing work (in particular work on implicit differentiation) I assert that this paper is closer to TorchDyn (as TorchDyn attempts to streamlines Neural ODE). The referenced NeurIPS 2019 paper is markedly different because it focuses on a specific type of problem (a subset of disciplined convex programs) and develops specialized methods (the grammar DPP) to solve this particular case. The authors will undoubtedly contest that their paper is closer to the NeurIPS 2019 paper in scope, but this can be discussed in the comment on unifying perspective below.
> >
> > ## On the unifying perspective
> >
> > In my initial review, I tacitly assumed that the contribution was that this paper greatly reduced the barrier of entry for working with implicit differentiation (with a well built Jax library). This is a worthwhile contribution, but I should've made my point more clear: the notion of the "unifying perspective" of implicit differentiation did not (and still does not) sit right with me. In particular, I remain unconvinced that the unifying perspective does anything more than just repackage existing methods. This is evidenced by how one incorporates existing solvers: this process requires a relatively deep understanding of the optimization method. For someone working on implicit differentiation (for say a different optimization class), the python package greatly simplifies their development pipeline, but this also means that they are aware of implicit differentiation which kind of defeats the point of this new perspective.
> >
> > I would also caution the authors against overclaiming that this view of implicit differentiation is entirely novel, as the methods described all depend on and cite the implicit function theorem (for example the OptNet paper cites "Dontchev, Asen L and Rockafellar, R Tyrrell. Implicit functions and solution mappings.").
> >
> > ## On Theorem
> >
> > As I mentioned in my initial review, the theorem is very distinct from the other contributions. Because of this, I judge the "theory" insights, in which aspects such as proof techniques matter. In particular, it should be noted that while the authors claim that this is an integral part of their paper, **this contribution is not even referenced in the abstract** and is only referenced once in the introduction. Overall, the theorem would be a fine addition to a paper that e.g. analyzes the empirical use of implicit differentiation over solver differentiation. But, this is not that paper.
> >
> > ## On Experiments
> >
> > I am not referring to the dimensionality of the data when I refer to the scale of the experiments; I am referring to downstream applications, motivation, and experimental analysis. For instance, consider the OptNet paper's test on TV denoising example. The task is well motivated by signal processing and has a neural network baseline. By comparison, I don't believe many of the tasks in this paper satisfy both of these (in fact only one experiment has a baseline and this is standard regularization). The DEQ results are in particular **much larger** scale than even the OptNet experiments (e.g. comparison with Transformers and SOTA language models at the time).

---

> > > ### Author Response · Authors · 2021-11-19
> > > **Response**
> > >
> > > Thank you very much for the clarifications.
> > >
> > > > For someone working on implicit differentiation (for say a different optimization class), the python package greatly simplifies their development pipeline, but this also means that they are aware of implicit differentiation which kind of defeats the point of this new perspective.
> > >
> > > We want to emphasize that **users do not need to be aware of implicit differentiation at all**. This is the whole point of our approach. Users just need to know the optimality conditions associated with their optimization problem and how to express them in Python, which is much easier.
> > >
> > > Analogously, deep learning practitioners can focus on expressing a loss function because deep learning frameworks have removed the need to compute gradients (both for people that know and those that don't know how to compute them). This is the goal of our paper for implicit differentiation.
> > >
> > > > I would also caution the authors against overclaiming that this view of implicit differentiation is entirely novel, as the methods described all depend on and cite the implicit function theorem (for example the OptNet paper cites "Dontchev, Asen L and Rockafellar, R Tyrrell. Implicit functions and solution mappings.").
> > >
> > > While the Optnet paper cites this reference, it does not mention the implicit function theorem. In Section 3, they differentiate KKT conditions manually in equations (5) and (6), using matrix differential calculus, citing (Magnus & Neudecker, 1988) as a reference. The connection with the implicit function theorem and root finding is not clear.
> > >
> > > In our approach, we reduce KKT conditions to root finding and use the implicit function theorem to differentiate the root, which is a different, albeit equivalent approach. We do believe the contribution of the Optnet paper is extremely valuable, but it is different, and, ultimately, less versatile than that proposed in our paper.
> > >
> > > >  this contribution is not even referenced in the abstract
> > >
> > > Indeed, we will add it to the abstract, following your comment. We will also better connect our theoretical result in the introduction.
> > >
> > > > the theorem would be a fine addition to a paper that e.g. analyzes the empirical use of implicit differentiation over solver differentiation
> > >
> > > We do have empirical comparison of unrolling vs. implicit differentiation in 3 out of 4 experiments.
> > >
> > > > I am not referring to the dimensionality of the data when I refer to the scale of the experiments; I am referring to downstream applications, motivation, and experimental analysis. For instance, consider the OptNet paper's test on TV denoising example. The task is well motivated by signal processing and has a neural network baseline. By comparison, I don't believe many of the tasks in this paper satisfy both of these (in fact only one experiment has a baseline and this is standard regularization). The DEQ results are in particular much larger scale than even the OptNet experiments (e.g. comparison with Transformers and SOTA language models at the time).
> > >
> > > The Optnet paper proposes a new optimization layer and the DEQ paper proposes a new architecture based on root finding. It is thus natural to compare them with other layers or architectures as a baseline. In our case, 3 experiments are bi-level optimization problems and 1 experiment is a sensitivity analysis. In our opinion, the natural baseline is unrolling.

---

> > > > ### Comment · Reviewer_h9WU · 2021-11-20
> > > > **Clarification**
> > > >
> > > > ## On Implicit Differentiation Researchers
> > > >
> > > > Apologies for being unclear and using a tautology. I was specifically referring to researchers extending implicit differentiation methods beyond the scope of $\mathbb{R}^n \times \mathbb{R}^d \to \mathbb{R}^n$ (such as implicit differentiation on non-Euclidean domains, using discrete conditions, and for computing higher order gradients) which is a heavy area of research for implicit differentiation. I do agree that the proposed method does significantly lower the barrier of entry for the existing domain (e.g. standard continuous optimization procedures on Euclidean space).
> > > >
> > > > ## On Novelty of Unifying Perspective
> > > >
> > > > I believe that, in OptNet, equations 5 and 6 (as well as an example of 7) are the invocation of the implicit function theorem. Furthermore, there are more papers than just OptNet that use the implicit function theorem (e.g. DEQs).  For example, in http://implicit-layers-tutorial.org/implicit_functions/, the organizers of the NeurIPS 2020 workshop on implicit differentiation specifically point out the implicit function theorem. This, in particular, is the exact same derivation used for the @custom_fixed_point method in the paper. Note that one of the organizers of this workshop is Zico Kolter, who is a coauthor on both the OptNet and DEQ paper as well as many other papers in this field.
> > > >
> > > > The overall point (beyond these two examples) is that the connections between the implicit function theorem and differentiable optimization are very, very well known. While it's fine to claim that the paper is the first to systemically extract the line in the derivation which uses the implicit function theorem and provide a package to better enable researchers to build models around this, **it would be an overstatment to say that this perspective is new since the implicit differentiation is used (implicitly or explicitly) in the field of implicit differentiation.** For example, in the abstract when the paper claim that this provides a "unified, efficient and modular approach for implicit differentiation", the word "unified" is by far an overclaim since the methods are already unified under the roof of implicit differentiation.
> > > >
> > > > This relates back to my point about novelty since the author response was that they introduced a "unified and easy-to-use mathematical framework for differentiating through a variety of optimality conditions". My point is that these conditions all use the implicit function theorem, and it is well known that they are using the implicit function theorem. Decoupling the implicit function theorem is useful, but this is not novel.
> > > >
> > > > ## On Theorem
> > > >
> > > > I don't think the paper's contribution is the evaluation of unrolling vs implicit differentiation, but rather the unifying perspective the authors present. Furthermore, I believe that all but 2 of the experiments do not line up with the narrative of the theorem. In particular, the main comparisons between unrolling and implicit differentiation show that unrolling is slower/requires more memory, and Figure 14 shows that unrolling doesn't seem to effect performance for that problem class. The only two figures that are related to Theorem 3 are Fig 2 and Fig 16.
> > > >
> > > > ## On Baselines
> > > >
> > > > I agree that many of the experiments do not have a natural baseline beyond unrolling. But, my point is that without a natural baseline the experiments are fundamentally implicit differentiation test cases, which are inherently more toy.

---

> > > > > ### Author Response · Authors · 2021-11-23
> > > > > **Comments taken into account**
> > > > >
> > > > > We thank the reviewer for the comments and for agreeing that our work significantly lowers the barrier of entry to implicit differentiation-based optimization.
> > > > >
> > > > > Since most of the reviewer’s objection against the paper seems to revolve around the use of the adjective “unified” in the abstract, we decided to remove it (this was the only occurrence of it in the whole paper anyway).
> > > > >
> > > > > Regarding the neurips tutorial mentioned by the reviewer, we already cite it and believe we provide a fair presentation of this work ("the tutorial requires the user to take care of low-level technical details and does not cover a large catalog of optimality condition mappings as we do”) as well as many others.
> > > > >
> > > > > Regarding Figure 14, we emphasize that the goal of this figure was to confirm that the faster runtimes of implicit differentiation compared to unrolling shown in Figure 3 were not at the cost of worse outer objective value (validation loss). We amended the caption of Figure 14 to clarify that.
> > > > >
> > > > > That said, we agree that Theorem 1, which is concerned with Jacobian error instead of outer objective value error, could be better connected with our experiments. To that end, we added Figure 15 in the revision, which does essentially the same thing as Figure 2 but for multiclass SVMs instead of ridge regression. We therefore now have three experiments confirming our theoretical result that implicit differentiation achieves smaller Jacobian error than unrolling, and believe that we have addressed your concern on this point.
> > > > >
> > > > > If you have any other recommendations, let us know.

---

### Official Review · Reviewer_Q3Lr · 2021-11-03

**Correctness:** 4
**Technical Novelty And Significance:** 3
**Empirical Novelty And Significance:** 3
**Recommendation:** 10
**Confidence:** 3

**Main Review:**

The paper proposes a modular and efficient framework along with its JAX implementation for the implicit differentiation of optimization problems. Firstly, the user defines the function F capturing the optimality conditions of the problem to be differentiated. Then the method frames the differentiation problem as a resolution of the linear system of equations (eq. 2) and combines implicit differentiation and autodiff of F to automatically differentiate the optimization problem.
The main novelty of the paper resides in the efficiency and modularity of the framework proposed to solve bi-level optimization problems. This framework allows to abstract away low-level details and significantly lowers the barrier to use implicit differentiation.
Bi-level optimization problems are ubiquitous in ML (hyperparameter optimization, meta-learning, NAS) and the framework proposed allows to address these problems efficiently.

The paper is well written and touches upon very relevant topics to the community. The framework, its implementation, and the theoretical insights of the Jacobian error are novel and useful contributions. Overall, the proposed method can positively impact the community since abstracting away low-level details of implicit differentiation allows opening new research directions. Therefore I strongly encourage accepting the paper (modulo minor comments below) and I think it should be highlighted at the conference.


COMMENTS:

1. Figure 1 proposes to the reader an illustrative example of the framework. Although the mathematical notation in the paper is well defined in a specific paragraph, the interpretation of the code functions/notation is left to the reader. For example, when you define “X_tr, y_tr = load_data()” the meaning of X_tr is left to the reader (might be X_transpose as well as X_train). Furthermore, an inexperienced reader with the numpy/jax notation might misunderstand the example proposed (functions jnp.eye and jnp.linalg.solve are not defined). I suggest you add line comments or a paragraph with the code notation.
2. In section 2.1 General Principle, the inner working of the method is proposed; however, it is difficult for me to untangle the novelties presented with respect to what is already known in the literature. Is the procedure of differentiating a root as a linear system part of the novelty? Or the main novelty resides in how to solve the linear system? Please clarify your contribution in this section.
3. The paper doesn’t contain an explicit definition of “Implicit Differentiation”. I think it would be beneficial to the reader to have a brief explicit definition of Implicit Differentiation.


MINOR TYPOS/GRAMMAR CORRECTION:
- “exiting implicit differentiation” -> “existing implicit differentiation”
- “we hope that this paper” -> “we hope this paper”
- “The derivation and implementation in these works is always case-by-case” -> “The derivation and implementation in these works are always case-by-case”
- “Often times” -> “Oftentimes”
- “this learned data set achieves small loss” -> “this learned data set achieves a small loss”


**Summary Of The Paper:**

The paper proposes a modular and efficient framework along with its JAX implementation for the implicit differentiation of optimization problems. The user defines the function F capturing the optimality conditions of the problem to be differentiated; then the framework combines implicit differentiation and autodiff of F to automatically differentiate the optimization problem. The proposed framework is labeled as efficient, since it doesn’t have to unroll the computational graph like in autodiff, and modular since it doesn’t require case-by-case mathematical derivation like in implicit differentiation.
The authors show that existing implicit differentiation methods can be instantiated in their framework. They provide and empirically validate new bounds on the Jacobian error when the optimization problem is only solved approximately.
The authors implemented four illustrative applications of their framework ( Hyperparameter Optimization Of Multiclass SVM; Dataset Distillation; Task-Driven Dictionary Learning; Sensitivity Analysis Of Molecular Dynamics).
Code and implementation in JAX are provided along with the paper.


**Summary Of The Review:**

The paper proposes a modular and efficient framework along with its JAX implementation for the implicit differentiation of optimization problems. This framework allows to abstract away low-level details and significantly lowers the barrier to use implicit differentiation. Bi-level optimization problems are ubiquitous in ML (hyperparameter optimization, meta-learning, NAS) and a procedure to automatically and efficiently tackle them is much needed by the community. I believe that this paper should be accepted and highlighted at the conference.

---

> ### Author Response · Authors · 2021-11-15
> **Response to reviewer Q3Lr**
>
> We thank the reviewer for the very positive review and useful comments.
>
> > Figure 1 proposes to the reader an illustrative example of the framework. Although the mathematical notation in the paper is well defined in a specific paragraph, the interpretation of the code functions/notation is left to the reader. For example, when you define “X_tr, y_tr = load_data()” the meaning of X_tr is left to the reader (might be X_transpose as well as X_train). Furthermore, an inexperienced reader with the numpy/jax notation might misunderstand the example proposed (functions jnp.eye and jnp.linalg.solve are not defined). I suggest you add line comments or a paragraph with the code notation.
>
> Thank you for the helpful suggestions. We added more detailed comments and replaced (X_tr, y_tr) by (X_train, y_train) in Figure 1.
>
> > In section 2.1 General Principle, the inner working of the method is proposed; however, it is difficult for me to untangle the novelties presented with respect to what is already known in the literature. Is the procedure of differentiating a root as a linear system part of the novelty? Or the main novelty resides in how to solve the linear system? Please clarify your contribution in this section.
>
> Our goal is  to (a) construct a new, modular approach using implicit differentiation, and (b) apply it to ML problems. Our contribution lies in decoupling the implicit differentiation scheme from the optimality condition specification. In existing works, both are intertwined, leading to a manual mathematical derivation and laborious implementation. A good example is Optnet (https://arxiv.org/abs/1703.00443), where the linear system is derived manually for the KKT conditions of a quadratic program (see their equations 5 and 6). In contrast, our approach advocates using JAX to differentiate KKT conditions automatically (see Figure 7).
>
> To answer your comment, we amended the following paragraph:
>
> Our approach is efficient as it can be added on top of any state-of-the-art solver and modular as the optimality condition specification is decoupled from the implicit differentiation mechanism. This contrasts with existing works, where the mathematical derivation and implementation are specific to each optimality condition.
>
> > The paper doesn’t contain an explicit definition of “Implicit Differentiation”. I think it would be beneficial to the reader to have a brief explicit definition of Implicit Differentiation.
>
> To make it clearer, we rephrased the introduction as follows:
>
> In recent years, two main approaches have been developed to circumvent this problem. The first one consists of unrolling the iterations of an optimization algorithm and using the final iteration as a proxy for the optimization problem solution. This allows to *explicitly* construct a computational graph relating the algorithm output to the inputs, on which autodiff can then be used transparently. [...]. In contrast, a second approach consists in *implicitly* relating an optimization problem solution to its inputs using optimality conditions. In a machine learning context, such implicit differentiation has been used for stationarity conditions, KKT conditions, and the proximal gradient fixed point.
>
> > Typos
>
> Thanks for pointing out these typos. We fixed all of them in the updated PDF version.

---

> > ### Comment · Reviewer_Q3Lr · 2021-11-20
> > **Clarifications**
> >
> > > Thank you for the helpful suggestions. We added more detailed comments and replaced (X_tr, y_tr) by (X_train, y_train) in Figure 1.
> >
> > Ok, I think the changes addressed the problem.
> >
> > > Our approach is efficient as it can be added on top of any state-of-the-art solver and modular as the optimality condition specification is decoupled from the implicit differentiation mechanism. This contrasts with existing works, where the mathematical derivation and implementation are specific to each optimality condition.
> >
> > I get the novelty of the paper is on proposing a new modular and efficient approach for implicit differentiation. My question was more on whether the content of section 2.1 that is called “General Principle” should be interpreted as a “Background” section where all the content proposed is known and not novel. My suggestion was to clarify your contribution separating what is known, ensuring that all the relevant related works are properly cited. Section 2.1 contains a lot of subparagraphs, maybe a strategy could be to add another section and split the current one.

---

### Decision · Program_Chairs · 2022-01-20

**Decision:**

Reject

**Comment:**

This paper was particularly discussed between the reviewers, the AC and SAC. A last minute reviewer was also called to clarify some issues raised, as one of the reviews never got into the system.

The paper was overall perceived as well written and well presented, and that the software contribution of implicit differentiation techniques is a nice asset for the community, especially its modularity.
The stability guaranty constitutes a nice (though straightforward) result providing a theoretical ground for the proposed approach.
Yet, the paper is often loose on mathematical justifications, in particular on minimal validity assumptions. Details on when the proposed framework could fail would be of interest, both on theoretical and practical parts. A discussion on the minimal assumptions required for validity of the approach should be highlighted more in the text.

Furthermore, the paper lacks discussions and comparisons with concurrent works,
for instance how would the framework compare with existing estimates for implicit differentiation or for unrolling. This could be improved along with providing more analysis on the implementation efficiency.
On the practical part, a high level description the software details would also be much beneficial.
A core discussion focused around what should be expected of this type of paper (i.e., "implementation issues, parallelization, software platforms, hardware" papers as suggested by Q3Lr)

A point of concern was the novelty aspects in the discussion phase was the novelty of the proposed framework: even if the contribution is the framework introduced, this is not new per se (the literature on implicit differentiation now contains a considerable amount of results and implementation examples).
The relevance of the work, both on theoretical and computational aspects, beyond the development of a computational library was found difficult to assess by several reviewers.
Overall, the reviewers judged the novelty and the paper's contribution more on the software side. Hence, a core discussion could focus on aspects expected for code oriented papers (i.e., implementation issues, parallelization, hardware, etc.).

Following the long discussion phase (more than 30 posts on OpenReview) and the aforementioned comments, the paper was rejected.

We encourage the authors to submit a revised version in a future conference or possibly to a software oriented journal, such as JMLR MLOSS or JOSS for instance.